# Shortcut Features as Top Eigenfunctions of NTK: A Linear Neural Network Case and More

**Jinwoo Lim**
Seoul National University
`jinwoolim8180@snu.ac.kr`

**Suhyun Kim**[*]
Kyung Hee University
`dr.suhyun.kim@gmail.com`

**Soo-Mook Moon**[*]
Seoul National University
`smoon@snu.ac.kr`

## Abstract

One of the chronic problems of deep-learning models is shortcut learning. In a case where the majority of training data are dominated by a certain feature, neural networks prefer to learn such a feature even if the feature is not generalizable outside the training set. Based on the framework of Neural Tangent Kernel (NTK), we analyzed the case of linear neural networks to derive some important properties of shortcut learning. We defined a "feature" of a neural network as an eigenfunction of NTK. Then, we found that shortcut features correspond to features with larger eigenvalues when the shortcuts stem from the imbalanced number of samples in the clustered distribution. We also showed that the features with larger eigenvalues still have a large influence on the neural network output even after training, due to data variances in the clusters. Such a preference for certain features remains even when a margin of a neural network output is controlled, which shows that the max-margin bias is not the only major reason for shortcut learning. These properties of linear neural networks are empirically extended for more complex neural networks as a two-layer fully-connected ReLU network and a ResNet-18.

## 1 Introduction

Based on various optimization algorithms, deep neural networks can learn rich features from data samples. Due to a success in advance of neural network architectures, neural networks can perform complex tasks as natural language processing. However, when neural networks are optimized with gradient-descent-based methods, there often exists a gap between a feature that is desired to learn and a feature that neural networks rely on. Rather than *core* features which can predict labels with a high accuracy, neural networks often rely on certain features called *shortcut features*. Shortcut features or *biased attributes* are features that show a strong but non-generalizable correlation to the ground-truth labels [8, 17]. Neural networks that learned shortcut features show minimum loss for the training set, but fail for data outside the training set. Such a phenomenon frequently occurs when neural networks are trained on a *biased dataset*, of which the majority of training data samples are influenced by the biased attributes [28]. This problem of shortcut learning is one of the most chronic problems of deep-learning models.

In this paper, we study the case where shortcut learning occurs due to the dominance of biased attributes on the majority of samples. We analyze the cause and effect of shortcut learning upon the framework of *Neural Tangent Kernel* (NTK) theory. We decompose the neural network outputs into eigenfunctions of NTK and measure the influence of each eigenfunction on neural network outputs. We study a simple case: training a linear neural network on a dataset of an almost-separable Gaussian mixture model, of which the majority of the samples are clustered by a certain feature. Under this assumption, we find interesting properties of shortcut learning.

---

[*]co-corresponding authors.

39th Conference on Neural Information Processing Systems (NeurIPS 2025).

First, shortcut features correspond to eigenfunctions with large eigenvalues. Due to larger eigenvalues, shortcut features are learned faster than other features, which has been empirically observed in previous works [14, 18, 23]. Second, shortcut features contribute more to the neural network output than other features after convergence. Therefore, while data samples with no shortcut features can also be learned with near-zero loss during training, such data outside the training set are still not well-predicted, which was also observed in previous works [20, 21]. In those previous works, the max-margin bias has been pointed out as a main underlying cause of shortcut learning. As an ablation study, we inspect the case where a margin of a neural network is penalized with a method called SD [19] or Marg-Ctrl [20]. We theoretically show that the decision boundary of a neural network can still be dominated by shortcut features even when the margin is controlled, hence the max-margin bias is not the only cause of shortcut learning. Another worth-noting thing is that an imbalance in feature contribution arises from the variance within groups of data samples.

Furthermore, we experimentally show that those properties of linear neural networks can be extended to the case of training complex neural networks such as ResNet-18 [9] on real-world datasets. For experiments, we introduce metrics to estimate how much a feature can predict the ground-truth labels, which we call *predictability*, and how much a feature is aligned to top eigenfunctions of NTK, which we call *availability*. We find that shortcut labels have lower predictability but higher availability in real-world data samples, which aligns with the results from linear neural networks.

Our main findings are as follows:

- For linear networks, features corresponding to clusters of larger weights have larger eigenvalues. Shortcut features which correspond to clusters of larger weights converge faster.
- Due to the data variance within each cluster, features corresponding to clusters of larger weights also have a larger influence on the output of a linear neural network. Thus, shortcut features which have larger eigenvalues also have a higher influence on the neural network output. Such a phenomenon can persist even when the margin of the neural networks is controlled with debiasing techniques as SD [19].
- We introduce metrics to estimate the predictability and the availability of real-world data samples. Shortcut features have a lower predictability but a higher availability.

## 2 Background

### 2.1 Notions of Neural Tangent Kernel

We briefly introduce some notions of Neural Tangent Kernel [11] in this section. The training set $\mathcal{D}$ is composed of input data $\mathcal{X} = \{x \in \mathbb{R}^d | (x, y) \in \mathcal{D}\}$ and ground-truth labels $\mathcal{Y} = \{y \in \mathbb{R}^k | (x, y) \in \mathcal{D}\}$. Here, we train a neural network $f : \mathcal{X} \to \mathcal{Y}$ of which the parameter is $\omega$. We assume a supervised learning setting where the neural network is trained with a loss function of $\mathcal{L} = \sum_{(x,y) \in \mathcal{D}} l(f(x, \omega), y)$. Most optimization algorithms in deep learning are based on gradient descent, which can be seen as an Euler method of first order solving an ODE called the *gradient flow*:

$$\dot{\omega} = -\eta \nabla_\omega \mathcal{L} = -\eta \nabla_\omega f(\mathcal{X}) \nabla_f \mathcal{L}. \tag{1}$$

From the equation above, we can describe the evolution of the output of the neural network as

$$\dot{f}(\mathcal{X}) = -\eta \nabla_\omega f(\mathcal{X})^\top \nabla_\omega f(\mathcal{X}) \nabla_f \mathcal{L} := -\eta K(\mathcal{X}, \mathcal{X}) \nabla_f \mathcal{L} \tag{2}$$

where $f(\mathcal{X}) \in \mathbb{R}^{k|\mathcal{D}| \times 1}$ denotes the concatenated outputs from all data from the training set. Here, the kernel $K(\mathcal{X}, \mathcal{X}) \in \mathbb{R}^{k|\mathcal{D}| \times k|\mathcal{D}|}$ is called *Neural Tangent Kernel* (NTK), which determines the convergence behaviour of neural network outputs.

As the width of the neural network goes to infinity, the network falls into a lazy-training regime [6] and the kernel remains constant during training. Thus, the kernel becomes deterministic at initialization which is noted as $K(\mathcal{X}, \mathcal{X}) \to K_0(\mathcal{X}, \mathcal{X})$. In this regime, for MSE loss $l(f, y) = \frac{1}{2} \|f - y\|_2^2$, the ODE has a rather simple solution shown in Lee et al. [15],

$$f(\mathcal{X}) = (I - e^{-\eta K_0 t}) \mathcal{Y}, \tag{3}$$

when the initial neural network output is zero. For a data point $x$, the neural network output is

$$f(x) = K_0(x, \mathcal{X}) K_0^{-1}(\mathcal{X}, \mathcal{X})(I - e^{-\eta K_0 t}) \mathcal{Y}. \tag{4}$$

Meanwhile, if the kernel can be eigendecomposed into $K_0(\mathcal{X}, \mathcal{X}) = \sum_i \lambda_i v_i v_i^\top$ where $\lambda_i$ is the $i$-th eigenvalue of $K_0$ with its corresponding eigenvector $v_i$, the output of the training set can also be decomposed and its convergence is expressed as

$$\langle v_i, \mathcal{Y} - f \rangle = e^{-\eta \lambda_i t} \langle v_i, \mathcal{Y} \rangle. \tag{5}$$

We can see that the convergence rate along the direction $v_i$ depends on the eigenvalue $\lambda_i$. The neural network learns directions with larger eigenvalues much faster than other directions, which is called the *spectral bias* [5].

Based on the eigendecomposition, it is possible to decompose the converged neural network output into eigenvectors of NTK as $f(x) = K_0(x, \mathcal{X}) \sum_i \lambda_i^{-1} v_i v_i^\top \mathcal{Y}$,

$$f(x) = \sum_i f^{(i)}(x); \quad f^{(i)}(x) = K_0(x, \mathcal{X}) \lambda_i^{-1} \langle v_i, \mathcal{Y} \rangle v_i \tag{6}$$

and $f^{(i)}$ was defined as a *feature* of input space in Tsilivis et al. [27]

## 2.2  Datasets

In this paper, we show our experimental observations on shortcut learning using biased datasets as depicted in Figure 1: Patched-MNIST [1], Colored-MNIST [13], Waterbirds [23], CelebA [16], and Dogs and Cats [12].

**Patched-MNIST.**  The task is to predict the digit: 0 - 4 are labelled as -1 and 5 - 9 are labelled as 1. The core feature is the shape of digits, while the non-core feature is the $3 \times 3$ patch at the corner of an image. 95% of digits labelled as -1 and only 5% of digits labelled as 1 were patched, and others were not patched for the training set. The ratio is balanced as 50% for the test set.

**Colored-MNIST.**  The task is to predict the digit: 0 - 4 are labelled as -1 and 5 - 9 are labelled as 1. The core feature is the shape of digits, while the non-core feature is the colour. 95% of digits labelled as -1 and only 5% of digits labelled as 1 were coloured red, and others were coloured blue for the training set. The ratio is balanced as 50% for the test set.

**Waterbirds.**  The task is to distinguish between waterbirds and land-birds. The core feature is the appearance of the bird, while the non-core feature is the background. 95% of waterbirds are in front of watery backgrounds, and 95% of landbirds are in front of land backgrounds for training set. The ratio is balanced as 50% for the test set.

**CelebA.**  The task is to determine if the hair colour of the figure is blonde or not. The core feature is the hair colour, while the non-core feature is other features such as gender. The ratio of the biased samples is naturally determined by the dataset and most blonde people are female.

**Dogs and Cats.**  The task is to distinguish dogs and cats. The core feature is the appearance of the animal, while the non-core feature is the colour of the animal. 95% of cats and only 5% of dogs are dark, while others are brightly coloured for the training set. The ratio is balanced as 50% for test set.

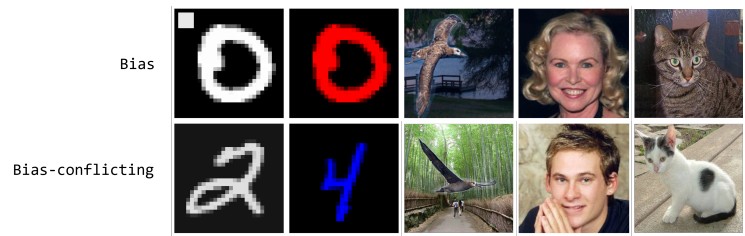

Figure 1: Datasets containing biased and core features: Patched-MNIST, Colored-MNIST, Waterbirds, CelebA, and Dogs and Cats.

# 3 Shortcuts on Gaussian Mixture Model

For the ease of analysis, we focus on a linear neural network with no activation function. The setting seems somewhat extreme and not applicable to real cases, but linear networks can still manifest important general attributes of neural networks such as the max-margin bias [20, 26] and kernel alignment [2, 4, 24], thus being worthy of research.

We also focus on a continuum limit where the number of discrete data samples is large and continuum approximation is possible. In such a setting, rather than a matrix, NTK is defined as a kernel function $k(\cdot, \cdot)$. Under a data distribution of $\mathcal{X} \sim \rho(x)$, matrix multiplication of NTK and a vector is defined as an application of an integral operator $T_K : L^2_\rho(\mathcal{X}) \to L^2_\rho(\mathcal{X})$ on a function, which is

$$T_K g(x) := \int_\mathcal{X} k(x, s) g(s) d\rho(s). \tag{7}$$

Instead of an eigenvector, a continuous setting allows us to define an eigenfunction $\phi(x)$ of a kernel operator as

$$\lambda \phi(x) = T_K \phi(x) = \int_\mathcal{X} k(x, s) \phi(s) d\rho(s) \tag{8}$$

where the eigenvalue is $\lambda$ and the eigenfunction is normalized to satisfy $\int_\mathcal{X} \phi^2(s) d\rho(s) = 1$.

For linear neural networks, the kernel function of NTK in the infinite-width limit for data $x$ and $y$ is proportional to $\langle x, y \rangle$ with a factor of the number of layers [2, 4]. Here, we define $k(x, y) := \langle x, y \rangle$ for analysis hereafter.

## 3.1 Problem formulation

Here, we define a simple setting of training data distribution. We assume a binary setting where all samples $x$ are labelled as $y \in \{-1, 1\}$. Data samples are clustered as a simple Gaussian mixture model based on the attributes of samples. The clusters are as follows: $B_{y,i} \sim \mathcal{N}(\mu_{B_{y,i}}, \sigma^2_{B_{y,i}})$ is the $i$-th cluster of samples with a biased attribute that were labelled as $y$, while $C_{y,i} \sim \mathcal{N}(\mu_{C_{y,i}}, \sigma^2_{C_{y,i}})$ is the $i$-th cluster of bias-conflicting samples that were labelled as $y$. Since we study the case where biased attributes dominate the majority of samples, in a mixture model, we assume that weights of $B_{y,\cdot}$ are larger than the weights of $C_{y,\cdot}$ as $\sum_i \pi_{B_{y,i}} \gg \sum_i \pi_{C_{y,i}}$. Though this setting studies a continuum limit, the assumed distribution is biased and it can approximate the finite samples of the biased training set if the number of training samples is large. Therefore, we assume that training data follow a distribution of

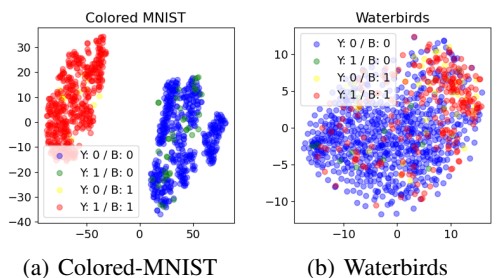

(a) Colored-MNIST  (b) Waterbirds

Figure 2: t-SNE visualization of 1000 input data samples in datasets. Each input is marked in color corresponding to its label - Y: ground-truth label, B: label from a shortcut feature.

$$p(x) = \sum_{y \in \{-1,1\}} [\sum_i \pi_{B_{y,i}} \mathcal{N}(\mu_{B_{y,i}}, \sigma^2_{B_{y,i}}) + \sum_j \pi_{C_{y,j}} \mathcal{N}(\mu_{C_{y,j}}, \sigma^2_{C_{y,j}})]. \tag{9}$$

Many previous studies have focused on the case of uniformly distributed data to simplify the analysis [5, 22, 29]. On the other hand, we assume a case of a clustered data distribution because the assumption on the uniform data distribution does not fully reflect real-world scenarios. In fact, in some cases, data are moderately *clustered* based on certain attributes of data samples. In Figure 2, it is possible to observe that Waterbirds and Colored-MNIST dataset, which are well-known datasets with shortcut features, are clustered by the biased attributes. This clustering is caused by the emergence of a shortcut feature, highlighting how strongly the biased attribute can influence the data distribution.

Another thing to note is that the data variance of each cluster is not really small. For the ease of analysis, some previous works [10] used an assumption of small covariance for the clustered distribution. However, in real cases, data samples are scattered and the variance is not small, which makes the small covariance assumption somewhat risky.

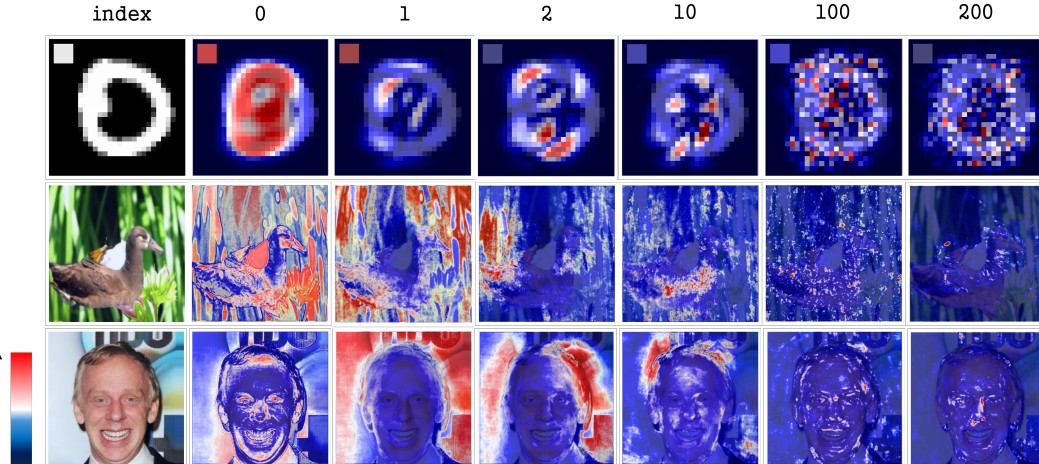

Figure 3: Original images and saliency maps from each feature of outputs from two-layer ReLU CNN networks. Saliency maps on the left side shows the spatial support of features with large eigenvalues, while saliency maps on the right side shows the spatial support of features with smaller eigenvalues. Indices indicate the ranks of the eigenvalues in terms of magnitude. A saliency map from the $i$-th index indicates the saliency map from a feature with the $(i + 1)$-th largest eigenvalue. Features with larger eigenvalues focus on biased attributes of samples, i.e., in CelebA, features with larger eigenvalues focus on the edges of the face, or the background of an image rather than the hair itself.

## 3.2 Eigenvalues of features

For kernel function $k(\cdot, \cdot)$, we can obtain a simple result about the eigenfunction and eigenvalue of the kernel.

**Proposition 3.1.** *Assume data $x \in \mathbb{R}^d$ in a Gaussian Mixture Model of $p(x) = \sum_{k=1}^{K} \pi_k \mathcal{N}(\mu_k, \sigma_k^2 I)$. The kernel $k(x, y) = \langle x, y \rangle$ has eigenfunctions $\phi_i$ and corresponding eigenvalues $\lambda_i$ as follows:*

$$\phi_i(x) = \begin{cases} x^\top v_i / c_i & \text{if } i = 1, \ldots, m \\ x^\top v_i^\perp / c_i & \text{otherwise} \end{cases} \tag{10}$$

$$\lambda_i = \begin{cases} \sum_{k=1}^{K} \pi_k \sigma_k^2 + a_i & \text{if } i = 1, \ldots, m \\ \sum_{k=1}^{K} \pi_k \sigma_k^2 & \text{otherwise} \end{cases} \tag{11}$$

*when $(\sum_{k=1}^{K} \pi_k \mu_k \mu_k^\top) v_i = a_i v_i$, $v_i^\perp$ is a vector perpendicular to $\mu_k$ for $k \in \{1, \ldots, K\}$, $m = rank(\sum_{k=1}^{K} \pi_k \mu_k \mu_k^\top)$, $\|v_i\| = 1$, $\|v_i^\perp\| = 1$ and $c_i = \sqrt{\lambda_i}$.*

The result suggests that the eigenvalue of the inner product with eigenvector $v_i$ of $\sum_{k=1}^{K} \pi_k \mu_k \mu_k^\top$ depends on the eigenvalue $a_i$. Since eigenvectors close to a cluster with a larger weight $\pi_k$ have larger eigenvalues, an inner product with a vector from a larger cluster has a larger eigenvalue and converges faster due to the spectral bias. As assumed in Section 3.1, $\pi_{B_{y,\cdot}}$ is much larger than $\pi_{C_{y,\cdot}}$. Therefore, by Proposition 3.1, we can expect that the inner products with data from these "shortcut clusters" will converge faster than others. Also, if means of clusters are close to each other, eigenvectors $v_i$ nearly orthogonal to the means $\mu_k$ have smaller eigenvalues $a_i$. In this case, features that help with distinguishing between clusters will have smaller eigenvalues and will converge more slowly. In a case where $\mu_{B_{y,\cdot}} \approx \mu_{C_{-y,\cdot}}$, it is harder to learn a neural network that distinguishes between biased samples and bias-conflicting samples.

We empirically extend this result for the case of a two-layer ReLU CNN model. For the experiment, we generated a saliency map [25] for each *feature* of a neural network from Equation 6 to highlight the region of interest for each feature. We computed the magnitude of the loss gradient with respect to the input, $\nabla_x \mathcal{L}(f_i, y)$, to measure the spatial support of each feature $f_i$. We use the same method as the implementation done in Tsilivis et al. [27].

Figure 3 shows the result. We can observe that features of larger eigenvalues rely on biased attributes. In Patched-MNIST, features with large eigenvalues focus on the patch of an image. In Waterbirds,

features of large eigenvalues focus on the background of an image. Also in CelebA, features of large eigenvalues focus more on the edges on the face, or the background of an image rather than the hair itself. This is the reason why neural networks learn shortcut features faster.

## 3.3 Features after convergence

In Proposition 3.1, we inspected the convergence rate of a neural network output during training. In this section, we inspect the influence of features on a neural network output after training. For linear neural networks, since the eigenfunctions are inner products, it is possible to dissect the neural network output into eigenfunctions of NTK. We will dissect the neural network output into a weighted sum of eigenfunctions of NTK, and investigate the magnitude of the weights to measure the influence of eigenfunctions on the neural network output. We assume that the neural network is trained with an MSE loss. Then we obtain the following result:

**Proposition 3.2.** *Assume data $x \in \mathbb{R}^d$ in a Gaussian Mixture Model of $p(x) = \sum_{k=1}^{K} \pi_k \mathcal{N}(\mu_k, \sigma_k^2 I)$. A binary label function $y(x) \in \{-1, 1\}$ nearly separates the mixture model that data from clusters with mean $\mu_c$ ($c \in \mathcal{C}$) are labelled as 1 and otherwise -1. When the linear neural network is optimized for $y(x)$ with the MSE loss and dissected into $f(x) = \sum_{k=1}^{m} w_k f_k(x)$ ($f_k(x) = x^\top v_k \propto \phi_k(x)$, $\|v_k\| = 1$),*

$$w_k = \frac{\sum_{j \in \mathcal{C}} \pi_j \mu_j^\top v_k - \sum_{j \in \mathcal{C}^c} \pi_j \mu_j^\top v_k}{\sum_{i=1}^{K} \pi_i \sigma_i^2 + v_k^\top (\sum_{i=1}^{K} \pi_i \mu_i \mu_i^\top) v_k} \tag{12}$$

*If $\mu_i \perp \mu_j$ for $i \neq j$ and $v_k = \mu_k / \|\mu_k\|$, then*

$$w_k = \begin{cases} \frac{\pi_k \|\mu_k\|_2}{\sum_{i=1}^{K} \pi_i \sigma_i^2 + \pi_k \|\mu_k\|_2^2} & \text{if } k \in \mathcal{C} \\ -\frac{\pi_k \|\mu_k\|_2}{\sum_{i=1}^{K} \pi_i \sigma_i^2 + \pi_k \|\mu_k\|_2^2} & \text{if } k \in \mathcal{C}^c \end{cases} \tag{13}$$

In the latter case, we assumed that means of clusters are orthogonal to each other so that a change in cluster weights $\pi_k$ would not change the eigenfunction up to a constant factor.

Since $w_k$ increases as $\pi_k$ increases from 0 to 1, eigenfunctions corresponding to clusters with larger weights have more influence on the converged neural network output. Considering from Proposition 3.1 that eigenfunctions corresponding to larger clusters have larger eigenvalues, it could be said that eigenfunctions with larger eigenvalues have more influence on the converged neural network output. As assumed in Section 3.1, data samples with biased attributes, $B_{y,\cdot}$, belong to a larger cluster ($\pi_{B_{y,\cdot}}$), so biased features have large eigenvalues and a large influence on the output at the same time. This shows that shortcut features, which have large eigenvalues, also have a large influence on the neural network output when a neural network is trained with the MSE loss.

One thing to note in Proposition 3.2 is that it not only applies to the lazy-training regime with constant NTK, but to general situations since this result is from the "optimal" function of neural networks in the distribution $p(x)$. Another thing to note is that such a dependency of $w_k$ on cluster weights $\pi_k$ originates from the existence of the variances, $\sigma_i$, among data in the clusters. If very small data variances were assumed as in [10], there would be no change in $w_k$ when there was a change in the distribution ($\pi_k$). The influence of shortcut features *after* convergence heavily depends not only on the weights of clusters, but on the variance of samples. Thus, a debiasing technique must consider the variance of samples if it aims to adjust the decision boundary of a neural network.

We empirically inspect the result of Proposition 3.2 with a toy example in a 2D space. In Figure 4, we trained a two-layer fully-connected ReLU network to classify four clusters of various weights and variances. The norms of the means of clusters were identical. The clusters on the x-axis become clusters of a shortcut feature. In Figure 4, the weight of clusters on the x-axis is denoted as $\pi_0$ and the decision boundary by the network was implicitly marked as a borderline between the regions of different colors. The results adhered to Proposition 3.2. When the variances in clusters were not small, the decision boundary was tilted towards the clusters with larger weights, and the inner products with samples from larger clusters had a larger influence on the decision boundary. However, when the variances of clusters were very small, the decision boundary did not depend on the weights of the clusters but solely on their positions, the same as the results from the linear kernel.

Another factor that affects the eigenvalues and the decision boundary is the norm of the mean vectors. Proposition 3.1 indicates that eigenvalues scale with both mixing weights ($\pi_k$) and the magnitude of

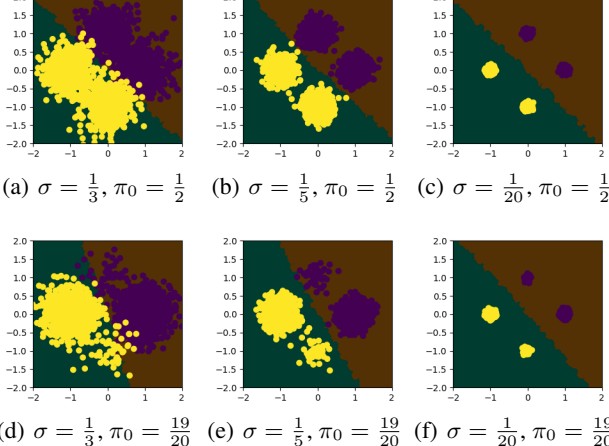

Figure 4: Classification of 4 clusters with a two-layer ReLU fully-connected network. The network was trained to classify yellow and purple data samples by their colors. The decision boundary by the network was implicitly marked as a borderline between the regions of different colors. For small variances, there is no change in borderline though there is a change of weights in the clusters. Only for large variances, there is a change in borderline when there is a change of cluster weights.

the mean vectors ($\|\mu_k\|_2$). In contrast, Proposition 3.2 implies that the contribution of a feature on the decision boundary ($w_k$) scales inversely with the magnitude of the mean vectors. While the scope of our paper is restricted to cases where the norms of the mean vectors are comparable (e.g. normalized), in scenarios where a cluster with a large weight has a very small-norm mean, its direction may no longer dominate the spectral bias, yet it can still exert a stronger influence on the decision boundary.

### 3.4 Discussion on the max-margin bias

Puli et al. [20] have also obtained a similar result under a case where linear networks are trained with the cross-entropy loss. In this work, the max-margin bias has been pointed out as a main cause of the imbalance in the feature contribution. In order to solve the problem, they inspected the case where the margin of the neural network is controlled and the parameter does not converge to the max-margin solution. Such control is done in a famous debiasing method known as *SD* [19] or *Marg-Ctrl* [20], which regularizes the parameter by adding a penalty on the norm of the neural network output. Thus a neural network is trained under a loss function below if the original loss is the cross-entropy loss:

$$l_{SD}(f(x), y) = \log(1 + \exp(-yf(x))) + \frac{\lambda}{2}|f(x)|^2 \tag{14}$$

In order to know if *"maximization"* of margins itself is the *only* problem, we inspect the case where the strength of the regularization is strong in SD.

**Corollary 3.3.** *Assume the linear neural network converges to $f_{SD}(x) = \sum_{k=1}^m (w_k)_{SD} f_k(x)$ ($f_k(x) = x^\top v_k \propto \phi_k(x)$, $\|v_k\| = 1$) when the network is trained with SD, and the linear neural network converges to $f_{MSE}(x) = \sum_{k=1}^m (w_k)_{MSE} f_k(x)$ when the network is trained with the MSE loss function. When the Lagrange multiplier in SD becomes infinite as $\lambda \to \infty$, then*

$$\lim_{\lambda \to \infty} \frac{(w_i)_{SD}}{(w_j)_{SD}} = \frac{(w_i)_{MSE}}{(w_j)_{MSE}} \tag{15}$$

*when $(w_j)_{SD} \neq 0$ and $(w_j)_{MSE} \neq 0$. $(w_i)_{MSE}/(w_j)_{MSE}$ does not change although SD is applied to the MSE loss function.*

First, if the original loss function is the MSE loss, the decision boundary does not change even when the regularization is applied. Second, when the regularization is strong, the decision boundary of SD converges to the decision boundary of a neural network trained with the MSE loss. Since a neural network trained with the MSE loss is affected by shortcut learning, this means that the neural network can still be affected by shortcut learning even when the margin is minimized. The max-margin bias is not the only cause of shortcut learning, but the shortcut bias arises from learning the label itself.

We empirically show Corollary 3.3 with a toy example in a 2D space in Figure 5. We trained a two-layer fully-connected ReLU network to classify four clusters with various loss functions. We used the MSE loss function with and without SD regularization, and the cross-entropy (CE) loss with and without SD. The regularization strength $\lambda$ was varied into two values: 0.1 and 1.0. The clusters on the x-axis become clusters of a shortcut feature.

As in Corollary 3.3, the decision boundary under the MSE loss did not change when SD was applied. The decision boundary under a CE loss changed when SD was applied. When the value of $\lambda$ was quite large as 1.0, the decision boundary converged to the one under the MSE loss.

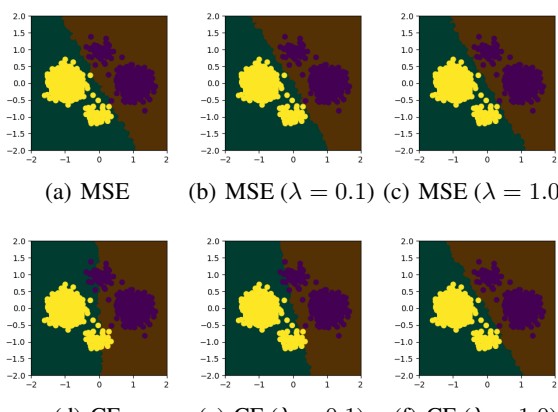

(a) MSE     (b) MSE ($\lambda = 0.1$) (c) MSE ($\lambda = 1.0$)

(d) CE     (e) CE ($\lambda = 0.1$)    (f) CE ($\lambda = 1.0$)

Figure 5: Classification of 4 clusters with a two-layer ReLU fully-connected network. The network was trained to classify yellow and purple data samples by their colors. SD was respectively applied to the MSE loss and CE loss. The decision boundary under SD converges to the one under the MSE loss.

## 4 Experiments

### 4.1 Predictability and availability

Here, we empirically extend theoretical results on linear neural networks to the case of more complex networks. For the experiments, we propose some new metrics. In Hermann et al. [10], some interesting methods were proposed to quantify how well a feature is aligned to the ground-truth and how much a feature is "easy" to learn for a neural network. These metrics are called *predictability* and *availability* of a feature, respectively. While they were based on *synthetic* datasets, we propose metrics for *realistic* datasets for experiments.

**Definition 4.1. Predictability** measures how well a feature $g$ is aligned with the ground-truth. Predictability is $\mathbf{y}^\top \mathbf{g}/|\mathcal{X}|$, which is $\int_\mathcal{X} y(x)g(x)d\rho(x)$ for a continuous case. $\mathbf{y} = [y(x) \text{ for } x \in \mathcal{X}]^\top$ is a vector of ground-truth labels, and $\mathbf{g} = [g(x) \text{ for } x \in \mathcal{X}]^\top$ is a vector of labels assigned by a certain feature $g(x)$.

**Definition 4.2. Availability** measures how "easy" it is to learn a label for a neural network. For a discrete case, availability is an alignment of label $\mathbf{g}$ to empirical NTK (eNTK) of $K(\mathcal{X}, \mathcal{X})$ [3],

$$A(K, \mathbf{g}) := \frac{\mathbf{g}^\top K(\mathcal{X}, \mathcal{X})\mathbf{g}}{\|\mathbf{g}\|_2^2 \|K(\mathcal{X}, \mathcal{X})\|_F} = \sum_i \frac{\lambda_i}{\sqrt{\sum_j \lambda_j^2}} \langle v_i, \frac{\mathbf{g}}{\|\mathbf{g}\|_2} \rangle^2 \qquad (16)$$

when $K(\mathcal{X}, \mathcal{X})$ can be eigendecomposed into $\sum_i \lambda_i v_i v_i^\top$.

Thus, availability measures how much top eigenspaces of NTK are aligned to a feature, which shows the convergence speed of a feature. By measuring availability of shortcut labels, we check if top eigenfunctions of NTK also have a larger influence after convergence as noted in Proposition 3.2.

### 4.2 Empirical results

Here, we measure the availability of shortcut labels on real-world datasets to compare it to the availability of the ground-truth labels. We used a pretrained ResNet-18 model for real-world datasets, which is a more complex network. Last-layer weights of both models were initialized as zeros and the bias parameter was removed. The model was trained with an SGD of a learning rate 0.001 and weight decay 0.0001. In the main paper, we only show the results from training with the CE loss, but we also show results from the MSE loss and SD in Appendix A.3.

**Finding a shortcut label.** Though a neural network is dependent on shortcut features, a label that a neural network prefers to learn can be actually different from the label solely from a shortcut feature.

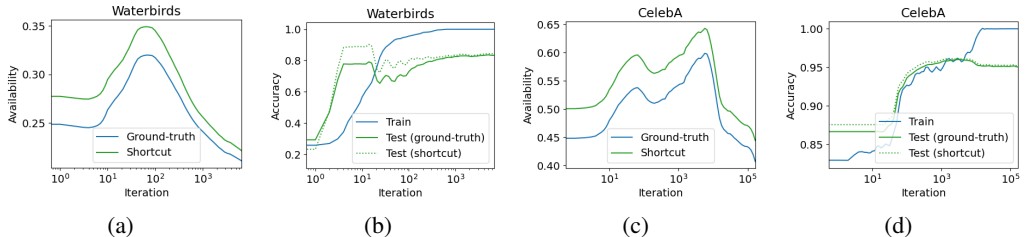

| (a) | (b) | (c) | (d) |

Figure 6: Availability and test accuracy of ground-truth labels and shortcut labels in two realistic datasets: Waterbirds [23] and CelebA [16]. The tested model was a pretrained ResNet-18. Availability was measured from 500 randomly sampled training data. Both the availability and the accuracy in the test sets were higher for shortcut labels than the ones for ground-truth labels. Smoothing was applied to the graphs for visual clarity. Results from five independent runs are provided in Appendix A.4.

By inspecting the predictions of a neural network on the test set, we manually found a *'shortcut label'* that a neural network actually learns. The shortcut labels from the datasets and their match rates to the test predictions are listed in Appendix A.1. We measured the availability of these manually-found shortcut labels for the experiments. Predictability of a shortcut label is lower than the ground-truth label by the way the dataset was composed.

**Shortcut label has higher availability.** The application of these metrics is shown in Figure 6. By the way the shortcut labels were found, the predictions of a neural network were closer to shortcut labels in real-world datasets. Meanwhile, the availability of these shortcut labels was larger than the availability of the ground-truth labels. Since the availability measures the alignment of a label to top eigenvectors of empirical NTK (eNTK), it is possible to say that the shortcut labels were more aligned with top eigenfunctions of eNTK. The empirical results were in line with the theoretical results of a linear neural network from Proposition 3.1 and Proposition 3.2. More results on synthetic datasets such as Colored-MNIST, Patched-MNIST, and Dogs vs Cats are in Appendix A.2 and A.5.

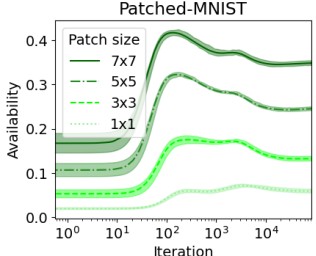

**Availability can reflect the strength of a shortcut.** This notion of availability is in line with the one from experiments of Hermann et al. [10]. Hermann et al. artificially manipulated the portion of images that were related to a certain feature to control availability. Similarly, in a linear kernel, the eigenvalues of shortcut features increase when the norms of means from shortcut clusters are large, so availability measures how much portion of input vectors is related to shortcut features. Thus, availability can reflect the strength of a shortcut.

Figure 7: The availability of shortcut labels in Patched-MNIST. The size of a patch is controlled to control the strength of the shortcut feature. The size varies from $1 \times 1$ to $7 \times 7$ pixels. The availability of the shortcut label was larger when the strength of the shortcut was larger.

We empirically extended this result to the case of other networks. In Patched-MNIST, we measured the availability of a two-layer ReLU fully-connected network varying the size of a patch in Patched-MNIST: $1 \times 1$, $3 \times 3$, $5 \times 5$, and $7 \times 7$ pixels. The result is shown in Figure 7. When a patch was large and the shortcut strength was strong, the availability of the shortcut label was large. This shows that availability can reflect the strength of a shortcut even in a case where the NTK kernel is not linear.

## 5 Conclusion

In this paper, we used the NTK framework to analyze shortcut learning caused by the imbalance in data distribution using a case of linear neural networks. We showed why shortcut features are learned quickly and are more influential on the neural network output than other features even after training. We discussed that the max-margin bias is not the only reason for the dominance of shortcut features on a neural network output. Finally, we proposed metrics to measure how a feature can predict ground-truth labels and how a feature can be easily learned by a network.

On the other hand, our work has limitations. First, our theoretical analysis is based on a simple case of training a linear neural network on a Gaussian Mixture model. Second, Proposition 3.1 is based on the NTK framework, which assumes an infinite width of the model. However, we empirically confirmed our results can be extended to general situations. Third, while Proposition 3.1 indicates that eigenvalues scale with the magnitude of the mean vectors, Proposition 3.2 implies that the contribution of a feature on the decision boundary scales inversely with the magnitude of the mean vectors. This could potentially limit the ability of using the NTK spectrum to assess the influence of shortcut bias after network convergence. Another limitation is that, as shown in Appendix A.7, availability might not detect certain shortcut labels if the ground-truth label is predominated by a stronger shortcut. These challenges are left for future works and we hope our work could be useful for future works that study the phenomenon of shortcut learning.

## 6 Acknowledgement

This work was supported in part by the National Research Foundation of Korea (NRF) grant funded by Korean Government [Ministry of Science and ICT (MSIT)] under Grant RS-2023-00208245, 50%; in part by the Institute of Information and Communications Technology Planning and Evaluation (IITP) grant funded by Korean Government (MSIT) under Grant 2021-0-00180, 10% and Grant 2021-0-00136, 10%; in part by the Information Technology Research Center (ITRC) support Program Supervised by IITP under Grant IITP-2021-0-01835, 20%; and in part by IITP under Artificial Intelligence Semiconductor Support Program to Nurture the Best Talents under Grant IITP-2023-RS-2023-00256081, 10%.

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

# A    Additional experiments

## A.1    Experimental details

**Shortcut labels.**    Though a neural network is dependent on shortcut features, a label that a neural network prefers to learn can be actually different from the label solely from a shortcut feature. For the experiments, by looking at the predictions of a neural network on the test set, we manually found a *'shortcut label'* that a neural network actually learns using both shortcut features and core features.

First, we grouped the samples into four groups: [Bias-aligned samples labelled as 1], [Bias-aligned samples labelled as -1], [Bias-conflicting samples labelled as 1], and [Bias-conflicting samples labelled as -1]. When the network converged, we observed how each group was predicted in the test sets and assigned a shortcut label to each group as the label predicted by the majority of samples in the group. For synthetic datasets, since the task is so simple that the neural network was able to learn core features as well, we composed the label solely from the shortcut features of the datasets.

**Colored-MNIST.**    If the colors of digits are green, then those samples are shortcut-labelled as 1. The rest are labelled as -1. Predictability of the shortcut label is 0.950.

**Patched-MNIST.**    If the digits are not patched, then those samples are shortcut-labelled as 1. The rest are labelled as -1. Predictability of the shortcut label is 0.950.

**Waterbirds.**    If the birds are waterbirds and the backgrounds are from water, then those samples are shortcut-labelled as 1. The rest are labelled as -1. Predictability of the shortcut label is 0.987.

**CelebA.**    If the person is blonde and is a woman, then the sample is shortcut-labelled as 1. The rest are labelled as -1. Predictability of the shortcut label is 0.991.

**Dogs vs Cats.**    If the sample contains a cat and a cat is dark-colored, then the sample is shortcut-labelled as -1. The rest are labelled as 1. Predictability of the shortcut label is 0.975.

| Ground-truth | Bias | Waterbirds | CelebA | Dogs vs Cats |
|---|---|---|---|---|
| 1 | 1 | $0.9268 \pm 0.0031$ | $0.8268 \pm 0.0117$ | $0.9384 \pm 0.0129$ |
| 1 | -1 | $0.4720 \pm 0.0137$ | $0.4044 \pm 0.0133$ | $0.9378 \pm 0.0112$ |
| -1 | 1 | $0.2396 \pm 0.0143$ | $0.0398 \pm 0.0028$ | $0.7742 \pm 0.0128$ |
| -1 | -1 | $0.0063 \pm 0.0007$ | $0.0081 \pm 0.0012$ | $0.0940 \pm 0.0169$ |

Table 1: The ratio of samples predicted as label 1 for each group in the datasets across 5 runs. 'Bias' denotes the label assigned solely from the shortcut feature. Shortcut label is assigned according to the label predicted by the majority of samples in each group.

**Experimental settings.**    Two models were trained for datasets: a two-layer ReLU FC network with the width of 256 for synthetic datasets, and a ResNet-18 pretrained with ImageNet for real-world datasets. The two-layer CNN model used for Figure 3 was composed of two convolutional layers with the kernel size of 3, padding of 1, ReLU activation, and a fully-connected layer. The models were trained with SGD of a learning rate 0.001 and weight decay 0.0001. The batch size was 128 for real-world datasets and 1000 for synthetic datasets. The models were learned with various loss functions: a cross-entropy loss with and without SD, and an MSE loss. For SD, the hyperparameter $\lambda$ was set 0.1 as in Puli et al. [20] The model was learned for 130 epochs on CelebA and 200 epochs on the rest of the datasets. Availability of ground-truth labels and shortcut labels was measured by 500 randomly sampled training data. The experiments were run on 2 RTX 3090 GPUs and AMD Ryzen Threadripper PRO 5955WX 16-Cores, for 6 hours for experiments with real-world data, 1 hour for synthetic datasets, and 12 hours for experiments with CelebA.

## A.2 Synthetic datasets

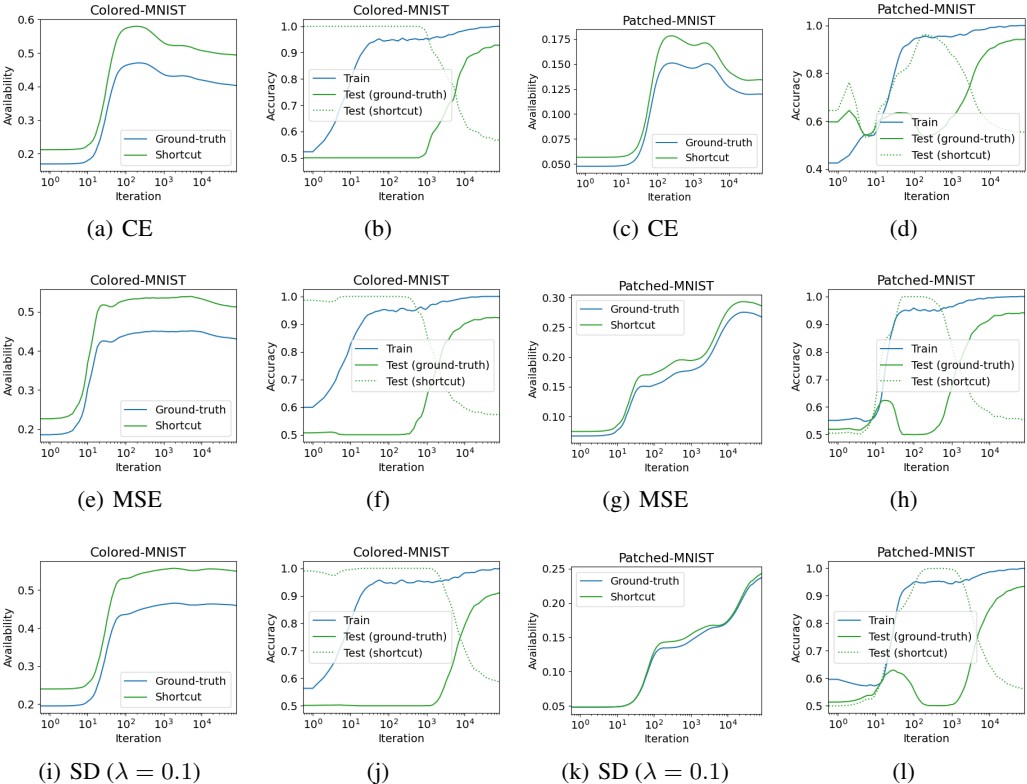

Figure 8: Availability and test accuracy of ground-truth labels and shortcut labels in two synthetic datasets: Colored-MNIST and Patched-MNIST. The tested model was a two-layer ReLU FC network. Availability was measured from 500 randomly sampled training data. Availability was higher for shortcut labels than the ones for ground-truth labels.

We measured the availability and test accuracies of ground-truth labels and shortcut labels in two synthetic datasets: Colored-MNIST and Patched-MNIST. The tested model was a two-layer ReLU FC network. Availability was measured from 500 randomly sampled training data.

As a result, availability of shortcut labels was larger than the availability of ground-truth labels, no matter what the loss function was. This also shows that shortcut labels are learned faster due to higher alignment to NTK, which extends the result of Proposition 3.1 to the case of a ReLU network. The model was able to partially learn the core feature as well and the test accuracy in shortcut labels was larger only in the early training phase. However, the test accuracies of the model ($\approx 0.92$ for Colored-MNIST under all losses, $\approx 0.94$ for Patched-MNIST under all losses) were lower than the test accuracy when the model was trained with the original MNIST ($> 0.98$). This implies that the model was affected by shortcut features and showed lower test accuracies.

Meanwhile, in such simple settings with a two-layer ReLU network, the trajectories of availability in SD resembled the one in MSE. This implies that the dynamics of SD could be mildly approximated by the dynamics of MSE if the regularization strength is high enough and the setting is simple. Another notable thing is that shortcut learning also occurred under the MSE loss and the SD loss, which means that shortcut learning can occur no matter if the margin is maximized or not. This shows that the max-margin bias is not the only main cause of shortcut learning even when the activation function is not linear.

## A.3   Real-world datasets

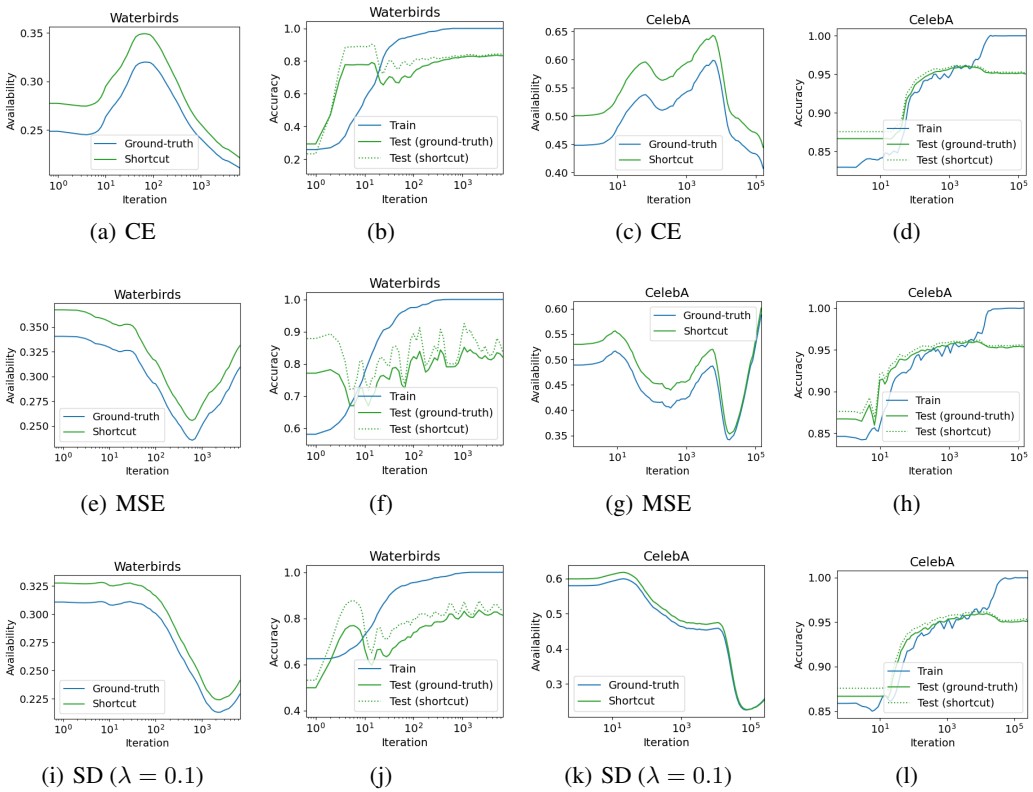

(a) CE  (b)  (c) CE  (d)

(e) MSE  (f)  (g) MSE  (h)

(i) SD ($\lambda = 0.1$)  (j)  (k) SD ($\lambda = 0.1$)  (l)

Figure 9: Availability and test accuracy of ground-truth labels and shortcut labels in two real-world datasets: Waterbirds and CelebA. The tested model was a pretrained ResNet-18. Availability was measured from 500 randomly sampled training data. Both the availability and the accuracy in the test sets were higher for shortcut labels than the ones for ground-truth labels.

Next, we measured the availability and test accuracies of ground-truth labels and shortcut labels in two real-world datasets: Waterbirds and CelebA. The tested model was a pretrained ResNet-18. Availability was measured from 500 randomly sampled training data.

As a result, availability of shortcut labels was larger than the availability of ground-truth labels. This also shows that shortcut labels are learned faster due to higher alignment to NTK. The test accuracy of shortcut labels ($\approx 0.83$ for Waterbirds under all losses, $\approx 0.953$ for CelebA under all losses) was larger than that of ground-truth labels ($\approx 0.82$ for Waterbirds under all losses, $\approx 0.951$ for CelebA under all losses), which shows that the output of a model was dominated by shortcut features. These empirical results extend the theoretical findings in the case of a linear neural network in Proposition 3.1 and 3.2.

A notable thing is that shortcut learning again occurred under the MSE loss and the SD loss, which shows that shortcut learning can occur no matter if the margin is maximized or not. This implies that the max-margin bias is not the only main cause of shortcut learning even when the model is complex.

## A.4 Availability from multiple runs

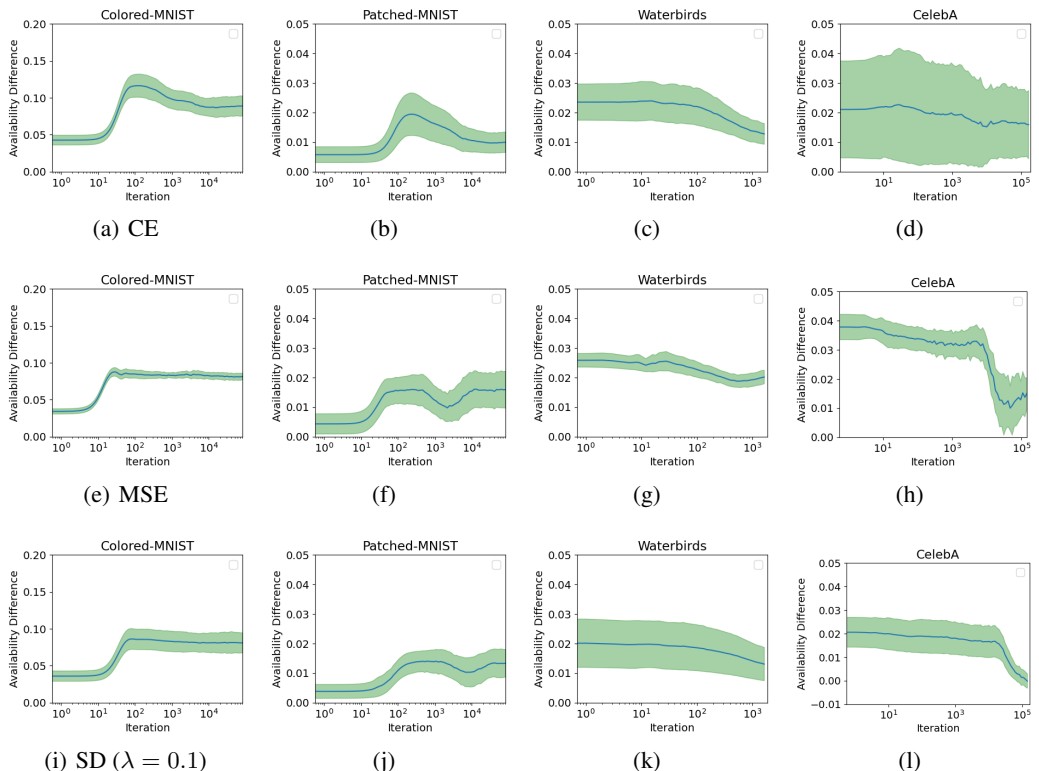

(a) CE      (b)      (c)      (d)

(e) MSE      (f)      (g)      (h)

(i) SD ($\lambda = 0.1$)      (j)      (k)      (l)

Figure 10: Availability of shortcut labels minus(-) availability of ground-truth labels in datasets: Colored-MNIST, Patched-MNIST, Waterbirds and CelebA. Availability was measured from 500 randomly sampled training data. The availability was higher for shortcut labels than the one for ground-truth labels.

Though the availability of a shortcut label was consistently higher than the availability of the ground-truth label, the values of availability varied due to the randomness in picking samples for measuring NTK. Thus, in Section A.3, we did not include error bars for the clarity of the graphs. Instead, we show the results from 5 runs in this section. Since the values of the availability varied, we show the availability of shortcut labels minus(-) availability of ground-truth labels. It is possible to observe that the availability was higher for shortcut labels than the one for ground-truth labels. We also show the results in a tabular form: we show availability of shortcut label - availability of ground-truth label at the 1000-th SGD update across other 5 runs. As a result, the availability of shortcut label was consistently higher than the availability of ground-truth label.

| Loss | Waterbirds | CelebA | Colored-MNIST | Patched-MNIST |
|------|------------|--------|---------------|---------------|
| CE | $0.0153 \pm 0.045$ | $0.0203 \pm 0.0162$ | $0.0977 \pm 0.0129$ | $0.0160 \pm 0.0061$ |
| MSE | $0.0182 \pm 0.0079$ | $0.0258 \pm 0.0086$ | $0.0832 \pm 0.0040$ | $0.0121 \pm 0.0048$ |
| SD | $0.0182 \pm 0.0064$ | $0.0164 \pm 0.0057$ | $0.0728 \pm 0.0220$ | $0.0140 \pm 0.0040$ |

Table 2: Availability of shortcut label - availability of ground-truth label at the 1000-th SGD update across 5 runs. Availability of shortcut label is higher than the availability of ground-truth label.

## A.5 Effect of pretraining

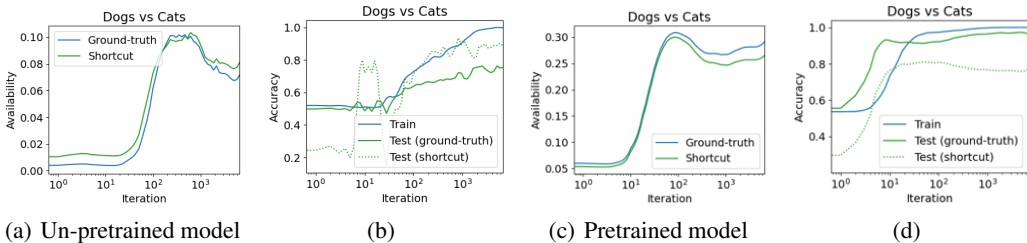

(a) Un-pretrained model    (b)    (c) Pretrained model    (d)

Figure 11: Availability and test accuracy of ground-truth labels and shortcut labels in Dogs vs Cats dataset. The tested model was an un-pretrained and a pretrained ResNet-18. Availability was measured from 500 randomly sampled training data. Since the pretrained model was already trained with samples of Dogs and Cats in ImageNet, availability was higher for ground-truth labels than the ones for shortcut labels when trained with a pretrained model.

In order to inspect the effect of pretraining, we trained ResNet-18 with both versions that were un-pretrained and pretrained with ImageNet [7]. Here we used Dogs vs Cats, which aims to distinguish dogs and cats but is spuriously correlated with colors of animals. ImageNet includes images of dogs and cats, therefore a pretrained ResNet-18 already contains information of dogs and cats. The model was trained with the cross-entropy loss.

As a result, availability of shortcut labels for an un-pretrained model was higher than the one of ground-truth labels in the early and late phase of training. Though this result aligns with previous experiments, availability of shortcut labels for an pretrained model was consistently lower than the one of ground-truth labels. The prediction of models in the test set was also heavily affected by pretraining. While an un-pretrained model was dependent on shortcut labels, a pretrained model was more dependent on ground-truth labels. This shows that the initial parameter of a model is a critical component of shortcut learning.

## A.6 Effect of correlation on availability

We measured the availability of the shortcut label varying the correlation between the shortcut label and the ground-truth label. Here, the correlation was controlled by manipulating the predictability of the shortcut label. As a result, the availability of a shortcut label was higher when the correlation between the labels was higher. This implies that it is easier to learn a shortcut label if the spurious correlation is stronger in the dataset.

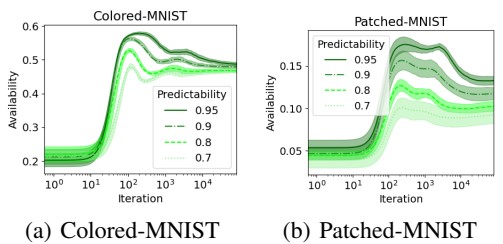

(a) Colored-MNIST    (b) Patched-MNIST

Figure 12: Availability of a model when the spurious correlation (predictability) is varied. When the label is more correlated, availability is higher.

## A.7 Limitation of availability due to multiple shortcuts

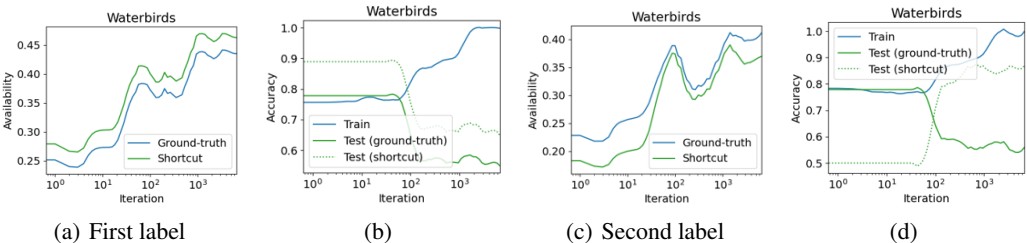

(a) First label      (b)      (c) Second label      (d)

Figure 13: Availability and test accuracy of ground-truth labels and shortcut labels in Waterbirds dataset. The tested model was an un-pretrained ResNet-18. Availability was measured from 500 randomly sampled training data.

We also trained an *un-pretrained* ResNet-18 on the modified Waterbirds dataset. Here, the sizes of the birds were 40% smaller than the original Waterbirds dataset so that the model would be more affected by the shortcut. Also, the un-pretrained model was less affected by the core feature and the shortcut label showed a different accuracy on the test sets. Here, the shortcut labels were composed in two ways. The first shortcut label is identical to the one from the experiment with a pretrained model. The second shortcut label is the label solely from the biased attribute: if the bird is on the watery background, then it is labelled as 1.

Both labels showed higher test accuracies than the ground-truth label, however, the speeds at which the labels were learned were different. The first shortcut label was learned faster, while the second label was learned slower than the ground-truth in the early phase of learning. Since the first label was learned quickly, the availability of the first label was larger than the ground-truth label. The second label was learned more slowly in the early phase, and the availability of the second label was consistently lower than the ground-truth. However, the test predictions of the model were closer to the second label.

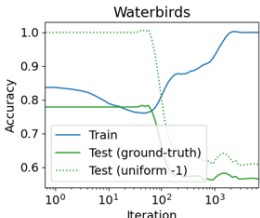

Figure 14: Comparing test predictions of model to a label uniformly composed as -1. The early predictions of a model is uniformly composed as -1, which is affected by the shortcut.

We found that this is because the model first learned to classify all the samples as land-birds due to the majority of label -1. The majority of the land-birds itself became *another shortcut*. Since the first label and the ground-truth were closer to the label uniformly composed as -1, the availability of the first label and the ground-truth label was actually larger than the one of the second label. If the ground-truth is affected by multiple shortcuts, then availability cannot detect the shortcut that is learned later than the other shortcut. We suspect that the difference in the predictability is the reason why the second label is more influential than the uniform label in the test predictions of an un-pretrained model. As implied in Section A.6, the predictability, or spurious correlation in other words, heavily affects the learning process. The predictability of the second label, 0.95, is much higher than the one of the uniform label, 0.78, and the test predictions were more affected by the second label. More studies are left for future work.

## B  Shortcuts in Gaussian Mixture Model

Now we provide the proofs for propositions in Section 3.

### B.1  Eigenvalues of features

Assume a Gaussian Mixture Model of input data $x \in \mathbb{R}^d$ below:

$$p(x) = \sum_{k=1}^{K} \pi_k \mathcal{N}(\mu_k, \Sigma_k) = \sum_{k=1}^{K} \frac{\pi_k}{(2\pi)^{d/2}|\Sigma_k|^{-1/2}} \exp\left(-\frac{1}{2}(s-\mu_k)^\top \Sigma_k^{-1}(s-\mu_k)\right) \tag{17}$$

For kernel function $k(\cdot, \cdot)$, an eigenfunction $\phi(x)$ must be an inner product with a vector,

$$\phi(x) = x^\top v \tag{18}$$

Since $\phi(x)$ is an eigenfunction for $k(\cdot, \cdot)$, the eigenfunction with eigenvalue $\lambda$ must satisfy

$$\lambda\phi(x) = \lambda x^\top v = T_K g(x) = \int_{\mathbb{R}^d} k(x, s)\phi(s)p(s)ds \tag{19}$$

$$= \int_{\mathbb{R}^d} x^\top s s^\top v \sum_{k=1}^{K} \frac{\pi_k}{(2\pi)^{d/2}|\Sigma_k|^{-1/2}} \exp\left(-\frac{1}{2}(s-\mu_k)^\top \Sigma_k^{-1}(s-\mu_k)\right)ds \tag{20}$$

$$= \sum_{k=1}^{K} \int_{\mathbb{R}^d} x^\top(s+\mu_k)(s+\mu_k)^\top v \frac{\pi_k}{(2\pi)^{d/2}|\Sigma_k|^{-1/2}} \exp\left(-\frac{1}{2}s^\top \Sigma_k^{-1}s\right)ds \tag{21}$$

$$= \sum_{k=1}^{K} \int_{\mathbb{R}^d} (x^\top s + x^\top \mu_k)(s^\top v + \mu_k^\top v) \frac{\pi_k}{(2\pi)^{d/2}|\Sigma_k|^{-1/2}} \exp\left(-\frac{1}{2}s^\top \Sigma_k^{-1}s\right)ds \tag{22}$$

$$= \sum_{k=1}^{K} \int_{\mathbb{R}^d} (x^\top s s^\top v + x^\top \mu_k v^\top s + \mu_k^\top v x^\top s + x^\top \mu_k \mu_k^\top v)$$

$$\frac{\pi_k}{(2\pi)^{d/2}|\Sigma_k|^{-1/2}} \exp\left(-\frac{1}{2}s^\top \Sigma_k^{-1}s\right)ds \tag{23}$$

$$= \sum_{k=1}^{K} x^\top \mathop{\mathbb{E}}_{s\sim\mathcal{N}(0,\Sigma_k)}[ss^\top]v + x^\top \mu_k \mu_k^\top v = \sum_{k=1}^{K} \pi_k x^\top \Sigma_k v + \pi_k x^\top \mu_k \mu_k^\top v \tag{24}$$

For simplicity, we take $\Sigma_k = \sigma_k^2 I$, then

$$\lambda\phi(x) = \lambda x^\top v = \sum_{k=1}^{K} \pi_k x^\top v \sigma_k^2 + \pi_k x^\top \mu_k \mu_k^\top v \tag{25}$$

To satisfy the equation above, $v$ should be an eigenvector of $\sum_{k=1}^{K} \pi_k \mu_k \mu_k^\top$ or perpendicular to $\mu_k$ for $k \in \{1, \dots, K\}$. If $\sum_{k=1}^{K} \pi_k \mu_k \mu_k^\top v = av$, then eigenvalue $\lambda$ becomes

$$\lambda = \sum_{k=1}^{K} \pi_k \sigma_k^2 + a. \tag{26}$$

When $v$ is perpendicular to $\mu_k$ for $k \in \{1, \dots, K\}$, then $a = 0$. Now we obtain the normalization factor $c$. Here we assume that eigenvector $v$ is normalized.

$$c^2 = \int_{\mathbb{R}^d} x^\top v x^\top v \sum_{k=1}^{K} \frac{\pi_k}{(2\pi\sigma_k^2)^{d/2}} \exp\left(-\frac{(x-\mu_k)^\top(x-\mu_k)}{2\sigma_k^2}\right) dx \tag{27}$$

$$= \sum_{k=1}^{K} \frac{\pi_k}{(2\pi\sigma_k^2)^{d/2}} \int_{\mathbb{R}^d} (x+\mu_k)^\top v (x+\mu_k)^\top v \exp\left(-\frac{x^\top x}{2\sigma_k^2}\right) dx \tag{28}$$

$$= \sum_{k=1}^{K} \frac{\pi_k}{(2\pi\sigma_k^2)^{d/2}} \int_{\mathbb{R}^d} (x^\top v v^\top x + v^\top \mu_k \mu_k^\top v) \exp\left(-\frac{x^\top x}{2\sigma_k^2}\right) dx \tag{29}$$

$$= \sum_{k=1}^{K} \pi_k \sigma_k^2 v^\top v + a = \lambda \tag{30}$$

Thus, the normalization factor $c = \sqrt{\lambda}$.

## B.2 Features after convergence

We first need to obtain the general form of the optimal function from Hermann et al. The process is identical to the process from Hermann et al. except that we do not decompose the function into normalized eigenfunctions but into $f(x) = \sum_{k=1}^{m} w_k f_k$ ($f_k(x) = x^\top v_k$). The function needs to minimize the following loss function:

$$\mathcal{L} = \int_{\mathcal{X}} \frac{1}{2}(f(x) - y(x))^2 d\rho(x) = \int_{\mathcal{X}} \frac{1}{2}\left(\sum_{k=1}^{m} w_k f_k(x) - y(x)\right)^2 d\rho(x) \tag{31}$$

We need to obtain $w_k$ that minimizes the loss function above, which satisfies

$$\frac{\partial}{\partial w_j}\mathcal{L} = \int_{\mathcal{X}} \left(\sum_{k=1}^{m} w_k f_k(x) - y(x)\right) f_j(x) d\rho(x) \tag{32}$$

$$= \left(\sum_{k=1}^{m} w_k \int_{\mathcal{X}} f_k(x) f_j(x) d\rho(x)\right) - \int_{\mathcal{X}} f_j(x) y(x) d\rho(x) = 0 \tag{33}$$

and it can be interpreted as

$$\sum_{k=1}^{m} w_k \int_{\mathcal{X}} f_j(x) f_k(x) d\rho(x) = \int_{\mathcal{X}} f_j(x) y(x) d\rho(x). \tag{34}$$

Thus, the weights are as follows:

$$w_k = (H^{-1}\mathbf{y})_k \tag{35}$$

where $H_{jk} = \int_{\mathcal{X}} f_j(x) f_k(x) d\rho(x)$ and $\mathbf{y}_j = \int_{\mathcal{X}} f_j(x) y(x) d\rho(x)$.

Now assume data of $\mathbb{R}^d$ in a Gaussian Mixture Model of $p(x) = \sum_{k=1}^{K} \pi_k \mathcal{N}(\mu_k, \sigma_k^2 I)$. A binary label function $y(x) \in \{-1, 1\}$ nearly separates the mixture model that data from clusters with mean $\mu_c$ ($c \in \mathcal{C}$) are labelled as 1. Under such assumptions, we obtain $H$ and $\mathbf{y}$.

$$H_{jk} = \int_{\mathcal{X}} f_j(x) f_k(x) d\rho(x)$$

$$= \int_{\mathbb{R}^d} x^\top v_j x^\top v_k \sum_{i=1}^{K} \frac{\pi_i}{(2\pi\sigma_i^2)^{d/2}} \exp\left(-\frac{(x-\mu_i)^\top(x-\mu_i)}{2\sigma_k^2}\right) dx \tag{36}$$

$$= \sum_{i=1}^{K} \frac{\pi_i}{(2\pi\sigma_i^2)^{d/2}} \int_{\mathbb{R}^d} (x^\top v_j v_k^\top x + v_j^\top \mu_i \mu_i^\top v_k) \exp\left(-\frac{x^\top x}{2\sigma_k^2}\right) dx \tag{37}$$

$$= \sum_{i=1}^{K} \pi_i \sigma_i^2 \delta_{jk} + v_j^\top \left(\sum_{i=1}^{K} \pi_i \mu_i \mu_i^\top\right) v_k \tag{38}$$

$$\mathbf{y}_j = \int_{\mathcal{X}} f_j(x) y(x) d\rho(x)$$

$$\approx \sum_{k \in \mathcal{C}} x^\top v_j \int_{\mathcal{X}} \frac{\pi_k}{(2\pi\sigma_k^2)^{d/2}} \exp\left(-\frac{(x-\mu_k)^\top(x-\mu_k)}{2\sigma_k^2}\right) dx$$

$$- \sum_{k \in \mathcal{C}^c} x^\top v_j \int_{\mathcal{X}} \frac{\pi_k}{(2\pi\sigma_k^2)^{d/2}} \exp\left(-\frac{(x-\mu_k)^\top(x-\mu_k)}{2\sigma_k^2}\right) dx \tag{39}$$

$$= \sum_{k \in \mathcal{C}} \pi_k \mu_k^\top v_j - \sum_{k \in \mathcal{C}^c} \pi_k \mu_k^\top v_j \tag{40}$$

Meanwhile, $f_j$ and $f_k$ are orthogonal to each other if $j \neq k$, therefore we can write as

$$w_k = \frac{\sum_{j \in \mathcal{C}} \pi_j \mu_j^\top v_k - \sum_{j \in \mathcal{C}^c} \pi_j \mu_j^\top v_k}{\sum_{i=1}^K \pi_i \sigma_i^2 + v_k^\top (\sum_{i=1}^K \pi_i \mu_i \mu_i^\top) v_k} \tag{41}$$

In the case where the means of clusters, $\mu_k$, are orthogonal to each other, the direction of $v_k$ does not change over the change of the weight $\pi_k$. $v_k$ always becomes $\mu_k/\|\mu_k\|_2$ or a unit vector orthogonal to the means. In this case, only diagonal elements of $H$ remain and the weights become

$$w_k = \begin{cases} \frac{\pi_k \|\mu_k\|_2}{\sum_{i=1}^K \pi_i \sigma_i^2 + \pi_k \|\mu_k\|_2^2} & \text{if } k \in \mathcal{C} \\ -\frac{\pi_k \|\mu_k\|_2}{\sum_{i=1}^K \pi_i \sigma_i^2 + \pi_k \|\mu_k\|_2^2} & \text{otherwise} \end{cases} \tag{42}$$

Due to the existence of $\sum_{k=1}^K \pi_k \sigma_k^2$, weight $w_k$ becomes larger when $\pi_k$ becomes larger. This means that features corresponding to heavier clusters have more influence on the output of a neural network.

### B.3  Margin controlling in CE loss

Here, we employ the same decomposition of a neural network output and inspect the case where the margin of the output is controlled. We use the same decomposition of a neural network output as $f(x) = \sum_{k=1}^m w_k f_k$ ($f_k(x) = x^\top v_k$). Since SD [19] is originally based on a cross-entropy loss, we first investigate the case where the regularization is done on a CE loss. Then, the function needs to minimize the following loss function:

$$\mathcal{L} = \int_{\mathcal{X}} [\log(1 + \exp(-y(x)f(x))) + \frac{\lambda}{2}|f(x)|^2] d\rho(x)$$

$$= \int_{\mathcal{X}} [\log(1 + \exp(-y(x)\sum_{k=1}^m w_k f_k(x))) + \frac{\lambda}{2}|\sum_{k=1}^m w_k f_k(x))|^2] d\rho(x) \tag{43}$$

We need to obtain $w_k$ that minimizes the loss function above, which satisfies

$$\frac{\partial}{\partial w_j} \mathcal{L} = \int_{\mathcal{X}} [\frac{-y(x)f_j(x)\exp(-y(x)f(x))}{1 + \exp(-y(x)f(x))} + \lambda \sum_{k=1}^m w_k f_k(x) f_j(x)] d\rho(x) = 0 \tag{44}$$

and it can be interpreted as

$$\lambda \int_{\mathcal{X}} \sum_{k=1}^m w_k f_k(x) f_j(x) d\rho(x) = \lambda w_j \int_{\mathcal{X}} f_j^2(x) d\rho(x) \tag{45}$$

$$= \int_{\mathcal{X}} \frac{\exp(-y(x)\sum_{k=1}^m w_k f_k(x))}{1 + \exp(-y(x)\sum_{k=1}^m w_k f_k(x))} y(x) f_j(x) d\rho(x) \tag{46}$$

since $f_k$ and $f_j$ are orthogonal to each other if $j \neq k$ in the first line.

Thus, the weights are as follows:

$$w_j = \frac{\int_{\mathcal{X}} \frac{\exp(-y(x)\sum_{k=1}^m w_k f_k(x))}{1 + \exp(-y(x)\sum_{k=1}^m w_k f_k(x))} y(x) f_j(x) d\rho(x)}{\lambda \int_{\mathcal{X}} f_j^2(x) d\rho(x)} \tag{47}$$

Here, for simplicity, we define a function $g(x, \lambda)$ as

$$g(x, \lambda) := \frac{\exp(-y(x) \sum_{k=1}^{m} w_k f_k(x))}{1 + \exp(-y(x) \sum_{k=1}^{m} w_k f_k(x))}. \tag{48}$$

Now we consider the case where $\lambda \to +\infty$. In this case, all weights $w_j$ go to 0. Though all weights go to zero, the ratio between those weights, which actually determines the decision boundary, can be different. First, we take the limit on the function $g(x, \lambda)$, then

$$\lim_{\lambda \to +\infty} g(x, \lambda) = \frac{1}{2}. \tag{49}$$

We take the limit on the ratio between $w_i$ and $w_j$ if $w_j \neq 0$,

$$\lim_{\lambda \to +\infty} \frac{w_i}{w_j} = \lim_{\lambda \to +\infty} \frac{\frac{\int_{\mathcal{X}} g(x,\lambda) y(x) f_i(x) d\rho(x)}{\int_{\mathcal{X}} f_i^2(x) d\rho(x)}}{\frac{\int_{\mathcal{X}} g(x,\lambda) y(x) f_j(x) d\rho(x)}{\int_{\mathcal{X}} f_j^2(x) d\rho(x)}} \tag{50}$$

$$= \frac{\int_{\mathcal{X}} f_j^2(x) d\rho(x)}{\int_{\mathcal{X}} f_i^2(x) d\rho(x)} \frac{\lim_{\lambda \to +\infty} \int_{\mathcal{X}} g(x,\lambda) y(x) f_i(x) d\rho(x)}{\lim_{\lambda \to +\infty} \int_{\mathcal{X}} g(x,\lambda) y(x) f_j(x) d\rho(x)} \tag{51}$$

$$= \frac{\int_{\mathcal{X}} f_j^2(x) d\rho(x)}{\int_{\mathcal{X}} f_i^2(x) d\rho(x)} \frac{\int_{\mathcal{X}} \lim_{\lambda \to +\infty} g(x,\lambda) y(x) f_i(x) d\rho(x)}{\int_{\mathcal{X}} \lim_{\lambda \to +\infty} g(x,\lambda) y(x) f_j(x) d\rho(x)} \tag{52}$$

$$= \frac{\int_{\mathcal{X}} f_j^2(x) d\rho(x)}{\int_{\mathcal{X}} f_i^2(x) d\rho(x)} \frac{\int_{\mathcal{X}} [\lim_{\lambda \to +\infty} g(x,\lambda)] y(x) f_i(x) d\rho(x)}{\int_{\mathcal{X}} [\lim_{\lambda \to +\infty} g(x,\lambda)] y(x) f_j(x) d\rho(x)} \tag{53}$$

$$= \frac{\int_{\mathcal{X}} f_j^2(x) d\rho(x)}{\int_{\mathcal{X}} f_i^2(x) d\rho(x)} \frac{\int_{\mathcal{X}} \frac{1}{2} y(x) f_i(x) d\rho(x)}{\int_{\mathcal{X}} \frac{1}{2} y(x) f_j(x) d\rho(x)} = \frac{\int_{\mathcal{X}} f_j^2(x) d\rho(x)}{\int_{\mathcal{X}} f_i^2(x) d\rho(x)} \frac{\int_{\mathcal{X}} y(x) f_i(x) d\rho(x)}{\int_{\mathcal{X}} y(x) f_j(x) d\rho(x)}. \tag{54}$$

Equation 52 is from the dominated convergence theorem, since

$$|\frac{\exp(-y(x) \sum_{k=1}^{m} w_k f_k(x))}{1 + \exp(-y(x) \sum_{k=1}^{m} w_k f_k(x))} y(x) f_i(x)| \leq |y(x) f_i(x)| \tag{55}$$

and the sequence is dominated by an integrable function.

If you look closer, Equation 54 is equivalent to the ratio of weights trained under an MSE loss. Therefore,

$$\lim_{\lambda \to \infty} \frac{(w_i)_{SD}}{(w_j)_{SD}} = \frac{(w_i)_{MSE}}{(w_j)_{MSE}} \tag{56}$$

When the original loss function is an MSE loss, we need to obtain $w_k$ that minimizes the loss function,

$$\mathcal{L} = \int_{\mathcal{X}} [\frac{1}{2}(f(x) - y(x))^2 + \frac{\lambda}{2}|f(x)|^2] d\rho(x)$$
$$= \int_{\mathcal{X}} [\frac{1}{2}(\sum_{k=1}^{m} w_k f_k(x) - y(x))^2 d\rho(x) + \frac{\lambda}{2}|\sum_{k=1}^{m} w_k f(x)|^2] d\rho(x) \tag{57}$$

We need to obtain $w_k$ that minimizes the loss function above, which satisfies

$$\frac{\partial}{\partial w_j} \mathcal{L} = \int_{\mathcal{X}} (\sum_{k=1}^{m} w_k f_k(x) - y(x) + \lambda \sum_{k=1}^{m} w_k f_k(x)) f_j(x) d\rho(x) \tag{58}$$

$$= (\sum_{k=1}^{m} (1 + \lambda) w_k \int_{\mathcal{X}} f_k(x) f_j(x) d\rho(x)) - \int_{\mathcal{X}} f_j(x) y(x) d\rho(x) = 0 \tag{59}$$

Therefore, when $w_j \neq 0$, the ratio between weights is

$$\frac{w_i}{w_j} = \frac{\frac{\int_{\mathcal{X}} y(x) f_i(x) d\rho(x)}{(1+\lambda) \int_{\mathcal{X}} f_i^2(x) d\rho(x)}}{\frac{\int_{\mathcal{X}} y(x) f_j(x) d\rho(x)}{(1+\lambda) \int_{\mathcal{X}} f_j^2(x) d\rho(x)}} = \frac{\int_{\mathcal{X}} f_j^2(x) d\rho(x)}{\int_{\mathcal{X}} f_i^2(x) d\rho(x)} \frac{\int_{\mathcal{X}} y(x) f_i(x) d\rho(x)}{\int_{\mathcal{X}} y(x) f_j(x) d\rho(x)} \tag{60}$$

which does not change though there is a change in the value of $\lambda$.

## C Broader impacts

The goal of our work is to understand the reason for shortcut learning and subsidiary phenomena accompanied by shortcut learning. As our work does not propose a new method and only aims to understand gradient-descent-like optimization algorithms, there is no downside risk of the research. The upside benefit would be to better understand the principles behind shortcut learning, which has been pointed out as socially problematic [10].

