# OpenReview forum: "Shortcut Features as Top Eigenfunctions of NTK: A Linear Neural Network Case and More"
_NeurIPS.cc/2025/Conference — NeurIPS 2025 spotlight_

### Official Review · Reviewer_o9y3 · 2025-07-02

**Clarity:** 3
**Significance:** 3
**Originality:** 3
**Rating:** 5
**Confidence:** 3

**Summary:**

The paper investigates shortcut learning in deep learning models, where networks overly rely on dominant but non-generalizable features in the training data. Using the Neural Tangent Kernel (NTK) framework, the authors analyze linear neural networks and define a “feature” as an eigenfunction of the NTK.
They find that shortcut features align with large NTK eigenvalues, particularly in settings with imbalanced or clustered data distributions. These dominant features significantly affect the model output even after training, suggesting that shortcut learning isn't solely due to max-margin biases. The insights are also shown to extend to nonlinear networks, like two-layer ReLU networks and ResNet-18.

**Questions:**

1. The theoretical analysis relies heavily on the NTK framework (infinite-width, lazy training). In practice, finite-width networks often exhibit feature learning.  Is there any empirical observations or theoretical intuitions about how the findings generalize to feature-learning regimes?

2. The shortcut labels in real-world datasets are manually constructed based on model predictions. How consistent are these across runs or architectures? Could a data-driven or automated method for shortcut label discovery be developed using availability scores?

**Ethical Concerns:**

["NO or VERY MINOR ethics concerns only"]

**Final Justification:**

The authors provided a bunch of empirical results provide more solid evidence on the claims. Also, the authors clarified most of the questions and concerns I have. Overall, I think it is a solid paper with all the extended results.

**Limitations:**

Theoretical results rely on idealized settings (linear networks, infinite width, Gaussian mixture data), which limits applicability.

The empirical results do not provide statistical significance analysis.

**Quality:**

3

**Strengths And Weaknesses:**

Strengths:

1. The paper presents a thorough theoretical analysis using the Neural Tangent Kernel (NTK) framework to dissect shortcut learning, starting from linear models and extending the insights to more complex architectures. It provides formal theorems and proofs (e.g., Theorem 3.1 and 3.2) that explain how shortcut features arise due to spectral properties of NTK and data distribution characteristics.

2. Introduces a clean interpretation of features as NTK eigenfunctions and relates shortcut features to eigenvalues, thereby linking shortcut learning to fundamental learning dynamics like spectral bias.

3. The paper empirically shows that findings from linear settings extend to two-layer ReLU networks and ResNet-18, increasing the real-world relevance of the theoretical insights.

Weaknesses:

1. Core theoretical results are derived under simplifying assumptions: linear networks, Gaussian mixture models, infinite-width limit. While extensions to complex models are attempted empirically, theoretical generality is limited.

2. The paper does not include error bars or statistical tests for empirical results. This undermines confidence in the generality of findings, especially on small or noisy datasets.

---

> ### Author Rebuttal · Authors · 2025-07-30
>
> Thank you very much for your constructive review!
> We first notify you that our equations 21, 22, 23 in Appendix B have typos: $\exp{(-\frac{1}{2} (s - \mu_k)^\top \Sigma^{-1}_k (s - \mu_k))}$ must be $\exp{(-\frac{1}{2} s^\top \Sigma^{-1}_k s)}$. We are sorry for the inconvenience.
>
> **1. The paper does not include error bars or statistical tests for empirical results. This undermines confidence in the generality of findings, especially on small or noisy datasets.**
>
> Though the availability of a shortcut label was consistently higher than the availability of the ground-truth label, the values of availability varied due to the randomness in picking samples for measuring NTK. Thus, we did not include error bars for the clarity of the graphs. But we will try to run important experiments such as Waterbirds and CelebA on CE loss to show you that our results are consistent.
> Though the values can vary due to randomness of picking training samples for measurement of NTK, the availability of shortcut labels was consistently higher than the one of ground-truth labels. Also, in the experiment measuring the strength of a shortcut (Fig. 7), a stronger shortcut always exhibited higher availability. The experiments below were run with CE loss. The availability was measured at the 1000-th iteration to observe the convergence behaviour in the early phase of training (which was also our focus in Fig. 6, 9, and 10)
> ## Waterbirds
> | | 1 | 2 | 3 | 4 | 5 |
> |--------------|--------|--------|--------|--------|--------|
> | Shortcut label | 0.3669 | 0.2700 | 0.3480 | 0.4049 | 0.2996 |
> | Ground-truth label | 0.3496 | 0.2576 | 0.3325 | 0.3827 | 0.2906 |
> ## CelebA
> | | 1 | 2 | 3 | 4 | 5 |
> |--------------|--------|--------|--------|--------|--------|
> | Shortcut label | 0.6300 | 0.5895 | 0.5886 | 0.5767 | 0.5958 |
> | Ground-truth label | 0.6257 | 0.5382 | 0.5707 | 0.5611 | 0.5833 |
> ## Colored-MNIST
> | | 1 | 2 | 3 | 4 | 5 |
> |--------------|--------|--------|--------|--------|--------|
> | Shortcut label | 0.5317 | 0.5360 | 0.5170 | 0.5347 | 0.5191 |
> | Ground-truth label | 0.4370 | 0.4248 | 0.4032 | 0.4472 | 0.4380 |
> ## Patched-MNIST
> | | 1 | 2 | 3 | 4 | 5 |
> |--------------|--------|--------|--------|--------|--------|
> | Shortcut label | 0.1628 | 0.1686 | 0.1751 | 0.1739 | 0.1684 |
> | Ground-truth label | 0.1555 | 0.1578 | 0.1533 | 0.1567 | 0.1454 |
>
> On the other hand, for the experiments measuring the strength of a shortcut in Patched-MNIST (Fig. 7), the variance of availability was small (across 5 runs):
> | Size of patch | 1x1 | 3x3 | 5x5 | 7x7 |
> |---------------|----------------|----------------|----------------|----------------|
> | Shortcut label | 0.0605 ± 0.0042 | 0.1698 ± 0.0049 | 0.2861 ± 0.0038 | 0.3721 ± 0.0085 |
>
> If given a chance to revise our paper, we will reflect your comment and include an error bar for this experiment.
> The experiments below were run with MSE loss.
> ## Waterbirds
> | | 1 | 2 | 3 | 4 | 5 |
> |--------------|--------|--------|--------|--------|--------|
> | Shortcut label | 0.2040 | 0.2342 | 0.1910 | 0.2406 | 0.2777 |
> | Ground-truth label | 0.1830 | 0.2078 | 0.1745 | 0.2173 | 0.2739 |
> ## CelebA
> | | 1 | 2 | 3 | 4 | 5 |
> |--------------|--------|--------|--------|--------|--------|
> | Shortcut label | 0.4466 | 0.4572 | 0.3853 | 0.3861 | 0.4841 |
> | Ground-truth label | 0.4140 | 0.4230 | 0.3715 | 0.3691 | 0.4527 |
> ## Colored-MNIST
> | | 1 | 2 | 3 | 4 | 5 |
> |--------------|--------|--------|--------|--------|--------|
> | Shortcut label | 0.5334 | 0.5529 | 0.5379 | 0.5529 | 0.5550 |
> | Ground-truth label | 0.4479 | 0.4670 | 0.4599 | 0.4742 | 0.4672 |
> ## Patched-MNIST
> | | 1 | 2 | 3 | 4 | 5 |
> |--------------|--------|--------|--------|--------|--------|
> | Shortcut label | 0.1695 | 0.1954 | 0.1528 | 0.1648 | 0.1718 |
> | Ground-truth label | 0.1544 | 0.1786 | 0.1431 | 0.1610 | 0.1566 |
> The experiments below were run with CE loss and SD regularization.
> ## Waterbirds
> | | 1 | 2 | 3 | 4 | 5 |
> |--------------------|--------|--------|--------|--------|--------|
> | Shortcut label | 0.2263 | 0.2237 | 0.1935 | 0.2318 | 0.2362 |
> | Ground-truth label | 0.2003 | 0.2122 | 0.1818 | 0.2155 | 0.2108 |
> ## CelebA
> | | 1 | 2 | 3 | 4 | 5 |
> |--------------------|--------|--------|--------|--------|--------|
> | Shortcut label | 0.4824 | 0.4093 | 0.4607 | 0.4243 | 0.4203 |
> | Ground-truth label | 0.4662 | 0.4015 | 0.4350 | 0.4068 | 0.4053 |
> ## Colored-MNIST
> | | 1 | 2 | 3 | 4 | 5 |
> |--------------------|--------|--------|--------|--------|--------|
> | Shortcut label | 0.5217 | 0.5533 | 0.5590 | 0.5293 | 0.5465 |
> | Ground-truth label | 0.4499 | 0.4548 | 0.4928 | 0.4934 | 0.4551 |
> ## Patched-MNIST
> | | 1 | 2 | 3 | 4 | 5 |
> |--------------------|--------|--------|--------|--------|--------|
> | Shortcut label | 0.1732 | 0.1542 | 0.1616 | 0.1572 | 0.1742 |
> | Ground-truth label | 0.1620 | 0.1453 | 0.1431 | 0.1447 | 0.1553 |
>
> We also answer your questions below:
>
> **1. The theoretical analysis relies heavily on the NTK framework (infinite-width, lazy training). In practice, finite-width networks often exhibit feature learning. Is there any empirical observations or theoretical intuitions about how the findings generalize to feature-learning regimes?**
>
> All the plots in Fig. 6, 9, and 10 are measurements from empirical NTK of finite-width networks. They were measured at >= 100 epoch (130 epoch for CelebA, 200 epoch for the rest). This empirically implies that our findings could be generalizable to feature-learning regimes. Also, one might extend our results with perturbation theory and dynamical mean-field theory, which might need more effort.
>
> **2. The shortcut labels in real-world datasets are manually constructed based on model predictions. How consistent are these across runs or architectures? Could a data-driven or automated method for shortcut label discovery be developed using availability scores?**
>
> The shortcut labels were consistent across runs. As in an example in Waterbirds, when the label was manually found, the match rate of the test predictions to the found label was **0.7819 ± 0.0018** (measured at > 100 epochs across 5 runs). On the other hand, when the label was composed solely from the shortcut feature, the match rate of the test predictions to the composed label was **0.7488 ± 0.0029** (across 5 runs) which was lower than the aforementioned one. In an example in CelebA, when the label was manually found, the match rate of the test predictions to the found label was **0.9527 ± 0.0003** (measured at > 100 epochs across 5 runs). On the other hand, when the label was the ground-truth label, the match rate of the test predictions to the ground-truth label was **0.9510 ± 0.0004** (across 5 runs) which was lower than the aforementioned one. In CelebA, we compared the match ratio of the shortcut label to the match ratio of the ground-truth label since the match rate to the label solely from the shortcut feature was much lower than the one to the ground-truth label. This shows that our shortcut label is reliable. But the shortcut labels were not consistent across parameters or architectures. For example, as shown in A.4, the shortcut labels might be different if the network is pretrained.
> And yes, it might be possible to construct a shortcut label from availability scores or from top eigenvectors of NTK (by constructing top eigenvectors of NTK as shortcut labels). But there is a limitation which was also noted in Appendix A.6: if there are multiple shortcuts in the datasets, it might not be possible to detect all the shortcut labels from the datasets.

---

> > ### Comment · Reviewer_o9y3 · 2025-08-06
> >
> > I thank the authors for their extended results and response! Those clarified my concerns and I would like to raise my score.

---

### Official Review · Reviewer_TcGE · 2025-07-02

**Clarity:** 3
**Significance:** 3
**Originality:** 3
**Rating:** 5
**Confidence:** 4

**Summary:**

The paper proposes an analysis of shortcut learning resulting from the dominance of biased attributes in the data for the case of linear networks using NTK theory, by tying the network features to the eigenfunctions of the NTK and measuring the influence of each eigenfunction to the neural network outputs.

The central parts of the theoretical argument are two theorems, formulated for the case of linear neural networks and datasets generated from mixtures of Gaussians with isotropic covariances:

**Theorem 3.1** quantitatively inspects the spectral bias by relating the convergence speed of each feature direction to the eigenvalues​ of the between-means covariance matrix M. It states that the eigenfunctions of the integral operator defined for the inner product kernel are linear projections on the eigenvectors of M and the eigenvalues decompose in the sum of average within-cluster variances, which is a constant term, and the eigenvalues associated with the eigenvectors of M. The theorem provides insight on why shortcut features (which, in the considered scenario are aligned with large cluster mixing probabilities), corresponding to larger M eigenvalues, are fitted faster: their kernel eigenvalues are large. In contrast, discriminative features that correspond to either small or similar clusters show a slower learning speed.

**Theorem 3.2** further shows that, under training with MSE loss, eigenfunctions of the kernel operator corresponding to clusters with larger mixing weights have a higher influence on the network output. Coupled with theorem 3.1, it states that shortcut features (which correspond to larger kernel eigenvalues) are fitted early and dominate the network predictions after convergence. One important note from the authors is that the impact of bias feature after convergence depend not only on the cluster mixing weights but also on the variances of the samples, highlighted empirically on a toy example.

The theoretical results and experiments on toy data additionally challenge the idea that maximal margin control (for example via SD) alone can eliminate the shortcut bias, by stating that the SD decision boundary converges to the MSE decision boundary in the infinite-limit of the regularization coefficient, which is affected by shortcut learning. Further, the authors provide metrics for the concepts of predictability (how well a feature is aligned with the ground truth) and availability (the alignment of the feature with the top eigenspace of NTK) and empirically apply them on real datasets.

**Questions:**

* Given the spectral bias analysis, I am curious whether you have thought about how the new spectral bias insights (and the metrics for availability and predictability) might inform designing further debiasing strategies?
* Since the theory assumes a static NTK (which holds for infinite-width limit), have you observed a particular width threshold beyond which the spectral bias observations tend to emerge in the experiments?
* As mentioned in the Weaknesses section, for the toy experiments highlighting theorems 3.1 and 3.2, has the norm of the cluster means been controlled in some way in order to observe its effect on the eigenvalue scales?
* Can some aspects from the theoretical argument be directly applied from linear NTK to other types of kernels?

**Ethical Concerns:**

["NO or VERY MINOR ethics concerns only"]

**Final Justification:**

Key updates in the rebuttal include the following points :
* reiterates where the GMM-data and constant-NTK premises are assumed in Theorems 3.1 and 3.2 and Corollary 3.3
* discussions on the case of small, high-variance clusters (discussion can be found in the replies to reviewers uhAz and jAtx)
* supplies preliminary confidence-interval experiments across multiple runs, strengthening the empirical claims (discussion can be found in the replies to reviewers jAtx and o9y3)
* tests on how measures of availability varies with network width, confirming that the availability of the shortcut label is consistently higher than the availability of the ground-truth.

However, I also emphasized in my discussion with the authors that, even if the empirical evaluation regarding Theorem 3.2 focuses exclusively on the role of the mixing weights, the paper should still discuss more clearly that eigenvalues are influenced by both the mixing weights and the magnitude of the mean vectors, a consequence deriving from Theorem 3.1. I recommended including this discussion such that the influence of mean-vector norms is not overlooked, as equal mean vectors constitutes another limiting assumption of the setup. I appreciate that the authors plan to detail this discussion for the final version.

Following the rebuttal, I have decided to increase my rating.

**Limitations:**

Yes

**Paper Formatting Concerns:**

No major paper formatting concerns

**Quality:**

3

**Strengths And Weaknesses:**

**Strengths**:
* The authors take an interesting perspective on shortcut features through the lens of NTK theory and advances the theoretical understanding
* The theoretical argument (restricted to the case of linear networks and GMM data) provides explicit forms for the eigenfunctions and weights as functions of the dataset mixing probabilities
* The results improve understanding on debiasing, by first challenging the idea that maximal margin alone cannot eliminate shortcut biases and by additionally highlighting the impact of within-cluster variance on both spectral bias and feature weights
* The experimental setup is extended beyond the limited regime considered in the theoretical argument (linear networks, GMM data) to real data and networks and empirically validates the spectral bias properties through newly introduced metrics for predictability and availability
* The results might potentially provide insights applicable for the study of interpretability, as shown in Figure 3 where saliency maps for directions with high and low eigenvalues are tied to human-interpretable features

**Weaknesses**:
* In the follow up analysis of the theorem 3.1, it is implied that eigenvalues depend primarily on the cluster mixing weights ("Since eigenvectors close to a cluster with a larger weight pk have larger eigenvalues, an inner product with a vector from a larger cluster has a larger eigenvalue and converges faster due to the spectral bias"). However, it follows from Theorem 3.1 that eigenvalues scale with both mixing weights and the magnitude of the mean vectors (concretely, with pk * ||uk||). Therefore, in scenarios where a cluster with a high mixing weight has a small-norm mean, its direction might not dominate the spectral bias anymore. I recommend clarifying this aspect in the analysis, as it potentially restricts the theoretical setup even further to datasets where the mean vectors have approximately equal norms, further drifting away from practice.
* Some of the limitations already pointed out by the authors: highly limiting regime in the theoretical analysis, with respect to data generation and class of models, as well as limitation of availability due to multiple shortcuts. While the empirical results show agreement with theory, the gap between the two is still considerable.
* Theory relies on static NTK assumption that only holds in the infinite-width limit

---

> ### Author Rebuttal · Authors · 2025-07-30
>
> Thank you very much for your constructive review!
> We first notify you that our equations 21, 22, 23 in Appendix B have typos: $\exp{(-\frac{1}{2} (s - \mu_k)^\top \Sigma^{-1}_k (s - \mu_k))}$ must be $\exp{(-\frac{1}{2} s^\top \Sigma^{-1}_k s)}$. We are sorry for the inconvenience.
>
> **1. I recommend clarifying this aspect in the analysis, as it potentially restricts the theoretical setup even further to datasets where the mean vectors have approximately equal norms, further drifting away from practice.**
>
> Firstly, both toy experiments in Fig. 4 and 5 were conducted with orthogonal clusters with means of norm 1. The norms of the means of clusters were identical and the effect of the norm was decoupled. While we were also aware of the effect of the norm of the mean, we did not elaborate on the subject since our focus was on the weight $\pi_k$. If given a chance to revise our paper, we will reflect your comment and emphasize more on the effect of norms of the means to the decision boundary of a neural network.
>
> **2. Theory relies on static NTK assumption that only holds in the infinite-width limit.**
>
> Theorem 3.1 assumes the static NTK assumption, which is the limitation of our work. However, it is known that the neural network can sometimes exhibit the lazy regime behaviour in the early phase of training, which is a focus of our analysis on the convergence speeds of features [3]. On the other hand, Theorem 3.2 and Corollary 3.3, which show the decision boundary of a network after convergence of the network, are not based on the constant NTK assumption.
>
> We also answer your questions below:
>
> **1. Given the spectral bias analysis, I am curious whether you have thought about how the new spectral bias insights (and the metrics for availability and predictability) might inform designing further debiasing strategies?**
>
> Our new spectral bias insights might aid finding possible shortcut labels inherent in the dataset. Since shortcut features are learned faster than core features, one can construct shortcut labels by looking at the predictions of the network in the early phase of training, which is already done in a famous debiasing method called LfF [1].
>
> **2. Since the theory assumes a static NTK (which holds for infinite-width limit), have you observed a particular width threshold beyond which the spectral bias observations tend to emerge in the experiments?**
>
> We could not particularly find a case where our observations were not valid. We show our results on Patched-MNIST and a two-layer ReLU FC network when the width of the network is small as 32 or 64: availability of the shortcut label was consistently higher than the availability of the ground-truth label.
> | Width | 32 | 64 | 256 | 512 | 1024 | 2048 |
> |-------------------|--------|--------|--------|--------|--------|--------|
> | Shortcut label | 0.1726 | 0.1625 | 0.1686 | 0.1684 | 0.1494 | 0.1138 |
> | Ground-truth label | 0.1637 | 0.1539 | 0.1578 | 0.1571 | 0.1430 | 0.1050 |
>
> **3. As mentioned in the Weaknesses section, for the toy experiments highlighting theorems 3.1 and 3.2, has the norm of the cluster means been controlled in some way in order to decouple its effect on the eigenvalue scales?**
>
> Both toy experiments in Fig. 4 and 5 were conducted with orthogonal clusters with means of norm 1. The norms of the means of clusters were identical and the effect of the norm was decoupled.
>
> **4. Can some aspects from the theoretical argument be directly applied from linear NTK to other types of kernels?**
>
> You cannot directly apply the results to NTK from a ReLU network, but in some cases where eigenfunctions of a ReLU NTK include linear functions [2], our results might be valid for linear eigenfunctions in those simple cases.
>
> [1] Nam, Junhyun, et al. "Learning from failure: De-biasing classifier from biased classifier." Advances in Neural Information Processing Systems 33 (2020): 20673-20684.
> [2] Hermann, Katherine L., et al. "On the foundations of shortcut learning." arXiv preprint arXiv:2310.16228 (2023).
> [3] Lyu, Kaifeng, et al. "Dichotomy of Early and Late Phase Implicit Biases Can Provably Induce Grokking." The Twelfth International Conference on Learning Representations.

---

> ### Comment · Reviewer_TcGE · 2025-08-05
>
> I thank the authors for their responses! In the light of the discussions, I have increased my score.
>
> Regarding control for the mean-vectors, although the experiment accounts for equal mean-vector norms, my original curiosity (in hindsight, unclearly formulated, apologies for that) was rather about systematically controlling the drift between mean-vector norms such that its effect on the network outputs is observed.
>
> While explicitly evaluating the effect of the mean-vector norms could more strongly highlight the theoretical-empirical link, I recognize that the focus of the experiment is on the mixing weights. I welcome and support the decision of the authors to emphasize more clearly on the effect of the mean-vector norms on the eigenvalues in an updated version.

---

### Official Review · Reviewer_jAtx · 2025-07-02

**Clarity:** 3
**Significance:** 2
**Originality:** 2
**Rating:** 5
**Confidence:** 1

**Summary:**

Introduces an NTK-based theory explaining why shortcut (spurious) features dominate training: clusters with higher sample weight π acquire larger NTK eigen-values, are fitted fastest, and retain the largest post-training influence.

Proves (Theorems 3.1–3.2) these properties for linear networks; then shows empirically that two-layer ReLU nets and ResNet-18 exhibit the same behaviour on Patched-/Colored-MNIST, Waterbirds, CelebA, Dogs vs Cats.

Defines predictability (alignment to ground-truth) and availability (alignment to top-NTK eigen-vectors) and demonstrates that shortcut labels have low predictability but high availability across five biased benchmarks.

Argues, both theoretically (Cor. 3.3) and with 2-D toy data, that margin-control methods (SD / Marg-Ctrl) cannot fully remove shortcut bias.

**Questions:**

Finite-width drift – Could you run a width-sweep (e.g. 256-2 k channels) on Patched-MNIST to show when NTK approximation breaks?

Statistical confidence – Please repeat each benchmark over ≥5 seeds and report mean ± s.d.; how stable is the availability ranking?

Shortcut-label sensitivity – How do availability curves change if shortcut labels are derived automatically (e.g. k-means on top eigen-projections) instead of manual inspection?

Variance term in Thm 3.2 – Provide a concrete bound showing when a small but high-variance cluster can outweigh a large low-variance one.

Late-epoch NTK – Have you measured availability using the empirical NTK after feature learning (e.g. epoch 50 ResNet-18) to see if shortcut alignment grows?

**Ethical Concerns:**

["NO or VERY MINOR ethics concerns only"]

**Final Justification:**

I appreciate the authors' efforts to address my concerns. I raised my score to 5.

**Limitations:**

Authors acknowledge linear-kernel and infinite-width assumptions but societal-impact section is brief; suggest expanding on potential mis-use of availability for data filtering.

**Quality:**

3

**Strengths And Weaknesses:**

**Quality**
Strength
1. Solid linear-kernel proofs, clearly laid out in Appendix B.
2. Comprehensive experiments (5 real, 2 synthetic datasets) validate theory.

Weakness

3. All curves come from single runs; no confidence intervals reported.

4. Manual construction of shortcut labels may inject subjectivity.

**Clarity**

Strength

1 Helpful visuals (Fig. 3 saliency, Fig. 4-5 decision boundaries) illuminate ideas.

Weakness

2 Main text is heavy with notation; many key plots relegated to appendix.

**Significance**

Strength

1. Bridges spectral bias and shortcut learning, highlighting cluster variance as a new factor.

Weakness

2. Results remain in infinite-width/lazy-training regime; impact on feature-learning settings uncertain.

**Originality**

Strength

1. Novel availability metric linking shortcut strength to NTK spectrum; challenges margin-bias narrative.

Weakness

2. Core insight “large eigen-values ⇒ faster learning” is rooted in prior spectral-bias work, limiting novelty.
3. Core contribution also overlaps with Learnability in the Context of NTK: because mathematically Tr K = ∑ λᵢ, their large-k(x,x) “easy” samples are exactly authors large-λ “shortcuts”.

---

> ### Author Rebuttal · Authors · 2025-07-30
>
> Thank you very much for your constructive review!
> We first notify you that our equations 21, 22, 23 in Appendix B have typos: $\exp{(-\frac{1}{2} (s - \mu_k)^\top \Sigma^{-1}_k (s - \mu_k))}$ must be $\exp{(-\frac{1}{2} s^\top \Sigma^{-1}_k s)}$. We are sorry for the inconvenience.
>
> **1. All curves come from single runs; no confidence intervals reported.**
>
> Though the availability of a shortcut label was consistently higher than the availability of the ground-truth label, the values of availability varied due to the randomness in picking samples for measuring NTK. Thus, we did not include error bars for the clarity of the graphs. Due to the limited time, we cannot run all experiments with multiple runs during our rebuttal period: but we will try to run important experiments such as Waterbirds and CelebA on CE loss to show you that our results are consistent. Though the values can vary due to the randomness of picking training samples for measurement of NTK, the availability of a shortcut label was consistently higher than that of ground-truth label. Also, in the experiment measuring the strength of a shortcut (Fig. 7), a stronger shortcut always exhibited higher availability. The experiments below were run with CE loss. The availability was measured at the 1000-th iteration to observe the convergence behaviour in the early phase of training (which was also our focus in Fig. 6, 9, and 10)
> ## Waterbirds
> | | 1 | 2 | 3 | 4 | 5 |
> |--------------|--------|--------|--------|--------|--------|
> | Shortcut label | 0.3669 | 0.2700 | 0.3480 | 0.4049 | 0.2996 |
> | Ground-truth label | 0.3496 | 0.2576 | 0.3325 | 0.3827 | 0.2906 |
> ## CelebA
> | | 1 | 2 | 3 | 4 | 5 |
> |--------------|--------|--------|--------|--------|--------|
> | Shortcut label | 0.6300 | 0.5895 | 0.5886 | 0.5767 | 0.5958 |
> | Ground-truth label | 0.6257 | 0.5382 | 0.5707 | 0.5611 | 0.5833 |
> ## Colored-MNIST
> | | 1 | 2 | 3 | 4 | 5 |
> |--------------|--------|--------|--------|--------|--------|
> | Shortcut label | 0.5317 | 0.5360 | 0.5170 | 0.5347 | 0.5191 |
> | Ground-truth label | 0.4370 | 0.4248 | 0.4032 | 0.4472 | 0.4380 |
> ## Patched-MNIST
> | | 1 | 2 | 3 | 4 | 5 |
> |--------------|--------|--------|--------|--------|--------|
> | Shortcut label | 0.1628 | 0.1686 | 0.1751 | 0.1739 | 0.1684 |
> | Ground-truth label | 0.1555 | 0.1578 | 0.1533 | 0.1567 | 0.1454 |
>
> On the other hand, for the experiments measuring the strength of a shortcut in Patched-MNIST (Fig. 7), the variance of availability was small (across 5 runs):
> | Size of patch | 1x1 | 3x3 | 5x5 | 7x7 |
> |---------------|----------------|----------------|----------------|----------------|
> | Shortcut label | 0.0605 ± 0.0042 | 0.1698 ± 0.0049 | 0.2861 ± 0.0038 | 0.3721 ± 0.0085 |
>
> If given a chance to revise our paper, we will reflect your comment and include an error bar for this experiment.
> The experiments below were run with MSE loss.
> ## Waterbirds
> | | 1 | 2 | 3 | 4 | 5 |
> |--------------|--------|--------|--------|--------|--------|
> | Shortcut label | 0.2040 | 0.2342 | 0.1910 | 0.2406 | 0.2777 |
> | Ground-truth label | 0.1830 | 0.2078 | 0.1745 | 0.2173 | 0.2739 |
> ## CelebA
> | | 1 | 2 | 3 | 4 | 5 |
> |--------------|--------|--------|--------|--------|--------|
> | Shortcut label | 0.4466 | 0.4572 | 0.3853 | 0.3861 | 0.4841 |
> | Ground-truth label | 0.4140 | 0.4230 | 0.3715 | 0.3691 | 0.4527 |
> ## Colored-MNIST
> | | 1 | 2 | 3 | 4 | 5 |
> |--------------|--------|--------|--------|--------|--------|
> | Shortcut label | 0.5334 | 0.5529 | 0.5379 | 0.5529 | 0.5550 |
> | Ground-truth label | 0.4479 | 0.4670 | 0.4599 | 0.4742 | 0.4672 |
> ## Patched-MNIST
> | | 1 | 2 | 3 | 4 | 5 |
> |--------------|--------|--------|--------|--------|--------|
> | Shortcut label | 0.1695 | 0.1954 | 0.1528 | 0.1648 | 0.1718 |
> | Ground-truth label | 0.1544 | 0.1786 | 0.1431 | 0.1610 | 0.1566 |
>
> **2. Manual construction of shortcut labels may inject subjectivity.**
>
> Though we manually constructed shortcut labels, the assignment of labels does not depend on our subjective decision: for real-world datasets, we grouped the samples into four groups: [Bias-aligned samples labelled as 1], [Bias-aligned samples labelled as -1], [Bias-conflicting samples labelled as 1], and [Bias-conflicting samples labelled as -1]. We observed how each group was predicted in the test sets and assigned a shortcut label to each group as **the predicted label of the majority of samples in the group** (if the majority of the group was predicted as 1, then the shortcut label becomes 1 for this group). In Fig. 6, 9, and 10, the test predictions of the network were closer to our constructed shortcut labels rather than the original ground-truth labels, which shows that our constructed shortcut labels are valid. More explanation is in Appendix A.1.
> This is purely based on the statistics of the predictions not our subjective decisions. As in an example in Waterbirds, when the label was manually found, the match rate of the test predictions to the found label was **0.7819 ± 0.0018** (measured at > 100 epochs across 5 runs). On the other hand, when the label was composed solely from the shortcut feature, the match rate of the test predictions to the composed label was **0.7488 ± 0.0029** (across 5 runs) which was lower than the aforementioned one. In an example in CelebA, when the label was manually found, the match rate of the test predictions to the found label was **0.9527 ± 0.0003** (measured at > 100 epochs across 5 runs). On the other hand, when the label was the ground-truth label, the match rate of the test predictions to the ground-truth label was **0.9510 ± 0.0004** (across 5 runs) which was lower than the aforementioned one. In CelebA, we compared the match ratio of the shortcut label to the match ratio of the ground-truth label since the match rate to the label solely from the shortcut feature was much lower than the one to the ground-truth label. This shows that our shortcut label is reliable.
>
> **3. Results remain in infinite-width/lazy-training regime; impact on feature-learning settings uncertain.**
>
> Theorem 3.1 assumes the infinite-width/lazy-training regime, which is the limitation of our work. However, it is known that the neural network can sometimes exhibit the lazy regime behaviour in the early phase of training, which is the reason our analysis focuses on the convergence speeds of features [1]. On the other hand, Theorem 3.2 and Corollary 3.3, which show the behaviour of shortcut features after convergence of the network, are not based on the constant NTK assumption.
>
> We also answer your questions below:
>
> **1. Finite-width drift – Could you run a width-sweep (e.g. 256-2 k channels) on Patched-MNIST to show when NTK approximation breaks?**
>
> We measured availability for two-layer ReLU FC networks with various widths: 32, 64, 256, 512, 1024, and 2048. Availability of the shortcut label was consistently higher than the availability of the ground-truth label.
> | Width | 32 | 64 | 256 | 512 | 1024 | 2048 |
> |-------------------|--------|--------|--------|--------|--------|--------|
> | Shortcut label | 0.1726 | 0.1625 | 0.1686 | 0.1684 | 0.1494 | 0.1138 |
> | Ground-truth label | 0.1637 | 0.1539 | 0.1578 | 0.1571 | 0.1430 | 0.1050 |
>
> **2. Statistical confidence – Please repeat each benchmark over ≥5 seeds and report mean ± s.d.; how stable is the availability ranking?**
>
> We showed the partial results above: though the value of the availability itself could be unstable, the availability of a shortcut label was consistently larger than the availability of the ground-truth label.
>
> **3. Shortcut-label sensitivity – How do availability curves change if shortcut labels are derived automatically (e.g. k-means on top eigen-projections) instead of manual inspection?**
>
> We are not sure if we understood your question correctly, but if you mean automatic derivation by composing the label from k-means clustering on data projected to principal components, then the availability varied across runs. We experimented on Waterbirds and the label was constructed from 2-means clustering on data projected to principal components. Below shows the result:
> ## Waterbirds
> | | 1 | 2 | 3 | 4 | 5 |
> |--------------------|--------|--------|--------|--------|--------|
> | Ground-truth label | 0.3631 | 0.3481 | 0.2902 | 0.3498 | 0.3466 |
> | K-means label | 0.0878 | 0.0836 | 0.0460 | 0.0419 | 0.0409 |
> We suspect that the labels from k-means clustering were far from the ground-truth label.
>
> **4. Variance term in Thm 3.2 – Provide a concrete bound showing when a small but high-variance cluster can outweigh a large low-variance one.**
>
> If the means of the clusters are orthogonal to each other and the norms of the means are identical, a small but high-variance cluster cannot outweigh a large low-variance one. The variance term $\sum^K_{i=1} \pi_i \sigma^2_i $ in $w_k$ includes the variance of all clusters and $w_k$ is not dependent only on the variance of a single cluster. If at least one cluster has a positive variance, then a large cluster always outweighs small clusters.
>
> **5. Late-epoch NTK – Have you measured availability using the empirical NTK after feature learning (e.g. epoch 50 ResNet-18) to see if shortcut alignment grows?**
>
> All the experiments in Fig. 6, 9, and 10 show the alignment of empirical NTK after epoch 100 (except CelebA that was run for 130 epochs, all the experiments were run for 200 epochs). The availability grew in experiments on synthetic datasets, but it did not in experiments on real-world datasets.
>
> [1] Lyu, Kaifeng, et al. "Dichotomy of Early and Late Phase Implicit Biases Can Provably Induce Grokking." The Twelfth International Conference on Learning Representations.

---

> > ### Comment · Reviewer_jAtx · 2025-08-09
> >
> > I appreciate the authors' efforts to address my concerns. I will raise my score.

---

### Official Review · Reviewer_uhAz · 2025-07-03

**Clarity:** 2
**Significance:** 2
**Originality:** 3
**Rating:** 5
**Confidence:** 3

**Summary:**

Propose a toy model where the problem of shortcut features can be investigated, using a mixture of two Gaussians with biased clusters (unbalanced probabilities) and considerable variance. Demonstrate, using an NTK setting in a linear network, that in this setup, shortcut features get learned earlier and influence the learned predictions. Furthermore, it demonstrates that a previously suggested measure to avoid shortcut features, controlling the classification margin, does not seem to alleviate the problem. Propose measures of availability and predictability to identify shortcut features in real-world datasets and demonstrate that they exhibit similar properties.

**Questions:**

1. Can you show that the Gaussian mixture model exhibits shortcut features without using the NTK analysis?
2. Can you quantitatively show the parameters of the GMM under which shortcut feature do or do not show, based on the NTK analysis?

**Ethical Concerns:**

["NO or VERY MINOR ethics concerns only"]

**Final Justification:**

The authors have properly answered my questions and clarified things I misunderstood, so I increased my score. The manuscript would still benefit from being updated to include clarifications, fix typos, and correct the experimental results, which contained a mistake.

**Limitations:**

yes

**Quality:**

3

**Strengths And Weaknesses:**

## Strengths
 1. A toy model where the problem of shortcut features can be reproduced and against which possible solutions can be tested.
 2. Use an existing theoretic framework (NTK) in a simplified manner to better understand the learned features which correspond to shortcut features.
 3. The predictability and availability metrics seem reasonable and useful.
 4. Test to see if a previously proposed mitigation for shortcut features work on the model, and found it does not.

## Weaknesses
 1. Unclear logical structure. The manuscript would benefit from a more layered structured, which makes clearer claims. Specifically, it is unclear which results depend on the Gaussian mixture model, which on the NTK analysis, and which do not. I think the structure of the logical argument is:
   ** Propose a model of biased Gaussian mixture to test shortcut features (with equations).
   ** Show when the model demonstrates shortcut features and when it does not (e.g., yes when biased larger than something and variance larger than something, show figure).
   **Apply a previously suggested method to understand the learned features in this model, demonstrating that they exist, are learned faster, and contribute to task performance.
   ** Show that a previously suggested solution for shortcut features, controlling prediction margin, is not helpful in this model.
   **Based on the analysis, suggest measures for identifying shortcut features, availability, and predictability.
   ** Replicate the results of real-world datasets, without the model's assumptions.
 2. The predictability and availability metrics seem reasonable, to the point that I would guess they already have a name or previous usage.
 3. The NTK analysis is relevant only in the lazy regime, which is not what happens when neural networks get trained on real-world samples. The derived results provide only an intuitive understanding that needs to be tested.
4. While it is note that "the data variance of each cluster is not really small", the contribution of this phenomenon is not well investigated. The point of Figure 4 is unclear.

---

> ### Author Rebuttal · Authors · 2025-07-30
>
> Thank you very much for your constructive review!
> We first notify you that our equations 21, 22, 23 in Appendix B have typos: $\exp{(-\frac{1}{2} (s - \mu_k)^\top \Sigma^{-1}_k (s - \mu_k))}$ must be $\exp{(-\frac{1}{2} s^\top \Sigma^{-1}_k s)}$. We are sorry for the inconvenience.
>
> **1. Unclear logical structure. The manuscript would benefit from a more layered structured, which makes clearer claims.**
>
> We are sorry for our unclear statements. Your statements you wrote about the logical argument are correct. For more information, we clarify our results as below:
> Theorem 3.1 and Theorem 3.2 are based on data following the Gaussian mixture model, while Corollary 3.3 does not necessarily need data to be on a Gaussian mixture model. Those three results need an assumption that the trained model is a linear neural network. Meanwhile, the toy experiments from Fig. 4, Fig. 5, and Fig. 7 are trained on a two-layer ReLU FC network. Saliency maps (Fig. 3) were generated from a two-layer ReLU CNN network and predictability and availability (Fig 6) were measured from a pretrained ResNet-18.
>
> **2. The predictability and availability metrics seem reasonable, to the point that I would guess they already have a name or previous usage.**
>
> Yes, especially the availability metric has been widely used to measure the alignment of NTK to the ground-truth label [1], but it has not been used to investigate the phenomenon of shortcut learning.
>
> **3. The NTK analysis is relevant only in the lazy regime, which is not what happens when neural networks get trained on real-world samples. The derived results provide only an intuitive understanding that needs to be tested.**
>
> Theorem 3.1 is relevant in the lazy regime and our analysis on the convergence speeds of features is dependent on the lazy regime assumption. We acknowledge that it is our limitation. However, it is known that the neural network can sometimes exhibit the lazy regime behaviour in the early phase of training, which is our main focus of our analysis on the convergence speeds of features [2]. Also, Theorem 3.2 and Corollary 3.3 do not need an assumption of lazy regime.
>
> **4. While it is note that "the data variance of each cluster is not really small", the contribution of this phenomenon is not well investigated. The point of Figure 4 is unclear.**
>
> The fact that the data variances of the clusters are not small is used in Theorem 3.2, which implies that shortcut features dominate the decision boundary when the variances are not zero. The point of Fig. 4 is to empirically prove Theorem 3.2 and show the effect of data variances on the decision boundary.
>
> We also answer your questions below:
>
> **1. Can you show that the Gaussian mixture model exhibits shortcut features without using the NTK analysis?**
>
> In Theorem 3.2, we show that the Gaussian mixture model exhibits shortcut features after convergence without using any NTK analysis. (Unfortunately, we currently do not provide an analysis on convergence speeds of features during convergence without NTK analysis.)
>
> **2. Can you quantitatively show the parameters of the GMM under which shortcut feature do or do not show, based on the NTK analysis?**
>
> If the data distribution follows the Gaussian mixture model of $p(x) = \sum^K_{k=1} \pi_k \mathcal{N}(\mu_k, \sigma^2_K I)$ and $\sigma_k > 0$, then the imbalance in the weights $\pi_k$ can trigger shortcut learning after convergence. Spectral bias can occur even when $\sigma_k = 0$.
>
>
> [1] Baratin, Aristide, et al. "Implicit regularization via neural feature alignment." International Conference on Artificial Intelligence and Statistics. PMLR, 2021.
>
> [2] Lyu, Kaifeng, et al. "Dichotomy of Early and Late Phase Implicit Biases Can Provably Induce Grokking." The Twelfth International Conference on Learning Representations.

---

> > ### Comment · Reviewer_uhAz · 2025-08-05
> > **Response to authors**
> >
> > Thank you for your clarifications!
> >
> > Your results seem interesting and well-supported. I think that improving the clarity and exposition (next time) would have a huge effect on your ability to explain your results to the reader. Meanwhile, I will increase my score.

---

### Author Response · Authors · 2025-08-01
**Notification on our Waterbirds dataset**

We further notify the reviewers that the images of the birds in our Waterbirds dataset are padded with a margin during synthesis, so the sizes of the birds in our dataset are 40% of those of the CUB-200 dataset. This description on the experimental setting was omitted in our paper. We are sorry for this omission, and if given a chance to revise our paper, we will add this description to the paper. On the other hand, when we experiment with no padding, the availability of the shortcut label is still consistently higher (across 5 runs) than the availability of the ground-truth label, as shown below (shortcut label was manually found again and did not change). So, the different image size does not affect the main argument of our paper about the higher availability of the shortcut label. If given a chance to revise our paper, we will include this experimental result as main plots to our paper.

| Waterbirds in CE loss | 1 | 2 | 3 | 4 | 5 |
|-----------------------|--------|--------|--------|--------|--------|
| Shortcut label | 0.2547 | 0.2751 | 0.2972 | 0.3390 | 0.2788 |
| Ground-truth label | 0.2397 | 0.2553 | 0.2859 | 0.3300 | 0.2654 |

| Waterbirds in MSE loss | 1 | 2 | 3 | 4 | 5 |
|-----------------------|--------|--------|--------|--------|--------|
| Shortcut label | 0.2462 | 0.2664 | 0.2325 | 0.2144 | 0.2486 |
| Ground-truth label | 0.2240 | 0.2513 | 0.2267 | 0.2006 | 0.2338 |

| Waterbirds in SD regularization | 1 | 2 | 3 | 4 | 5 |
|-----------------------|--------|--------|--------|--------|--------|
| Shortcut label | 0.2005 | 0.2444 | 0.2330 | 0.2678 | 0.2534 |
| Ground-truth label | 0.1963 | 0.2212 | 0.2166 | 0.2449 | 0.2371 |

---

> ### Comment · Area_Chair_tXKu · 2025-08-06
>
> Thanks for self-reporting the error. Seems like an honest mistake, and, importantly, does not seem to affect the main claims in the paper.
>
> It goes without saying but if your paper were accepted, the final version must be updated to reflect the corrected and clarified results presented during the review. I would be explicitly looking out for it. Note, of course, that the discussions are still ongoing, and no decisions have been made.

---

### Note · Authors · 2025-08-13

For more information, we clarify our results as below: Theorem 3.1 and Theorem 3.2 are based on data following the Gaussian mixture model, while Corollary 3.3 does not necessarily need data to be on a Gaussian mixture model. Those three results need an assumption that the trained model is a linear neural network. Meanwhile, the toy experiments from Fig. 4, Fig. 5, and Fig. 7 are trained on a two-layer ReLU FC network with the width of 256. Saliency maps (Fig. 3) were generated from a two-layer ReLU CNN network, and predictability and availability (Fig 6) were measured from a pretrained ResNet-18. The two-layer CNN model used for Fig. 3 was composed of two convolutional layers with the kernel size of 3, padding of 1, ReLU activation, and a last fully-connected layer.

Also, for Waterbirds experiment with no padding, the match rate of the test predictions to the manually-found shortcut label was 0.8435 ± 0.0068 (measured at > 100 epochs across 5 runs). On the other hand, when the label was composed solely from the shortcut feature, the match rate of the test predictions to the composed label was 0.6415 ± 0.0046 (across 5 runs) which was lower than the aforementioned one.

---

### Decision · Program_Chairs · 2025-09-17

**Decision:**

Accept (spotlight)

**Comment:**

The paper formally connects the widely observed phenomenon of shortcut learning to spectral bias, proposing that shortcut features correspond to NTK eigenfunctions with the largest eigenvalues.  The authors supported their theory with extensive experiments across a wide range of datasets and models. The work also argues that max-margin bias is not the only cause of shortcut learning, and how that the phenomenon persists even when the network's margin is controlled. Further, the authors demonstrate that data variance within clusters also plays an important role in learning shortcut features.

Originally, the results were reported for a single run, but during the rebuttal the authors extended the results and repeated the experiments for more runs. Another concern noted by the reviewers is that the theory relies on a linear network in an infinite-width regime. To address this concern, the authors showed that their findings empirically hold across different architecture sizes.